# YourBench: Easy Custom Evaluation Sets for Everyone

**Sumuk Shashidhar**[1,2*] **Clémentine Fourier**[1*] **Alina Lozovskia**[1]
**Thomas Wolf**[1] **Gokhan Tur**[2] **Dilek Hakkani-Tür**[2]

[1] 😊 HuggingFace   [2] 🟧 UIUC

sumuks2@illinois.edu   clementine@huggingface.co

## Abstract

Large language models (LLMs) have rapidly outpaced traditional evaluation methodologies, with static benchmarks suffering from saturation, contamination, and domain-specificity limitations while human evaluation remains prohibitively expensive. We present **YourBench**, an open-source framework that transforms this evaluation paradigm by enabling automated generation of reliable, contamination-free benchmarks directly from user-provided documents without human annotation. To validate our approach, we successfully reproduce the challenging MMLU-Pro benchmark across 86 models spanning 400M to 405B parameters, achieving remarkable Pearson correlations of 0.91-0.99 while generating entirely novel questions for under $15 per model. This demonstrates that dynamically generated evaluations can match the discriminative power of expert-curated benchmarks while eliminating contamination risks. YourBench enables researchers to create domain-specific benchmarks in minutes rather than months. We demonstrate applications in agriculture, personalized education, and RAG training that were previously infeasible. By releasing the YourBench library, TEMPORA-0325 dataset, 150K+ generated QA pairs, and all evaluation traces, we provide the community with a practical solution to the challenge of keeping pace with rapidly evolving model capabilities.

## 1 Introduction

The rapid evolution of large language models (LLMs) continually outpaces traditional evaluation methodologies. Static benchmarks, foundational to earlier progress, now face critical issues: they quickly saturate, are susceptible to training data contamination, become temporally irrelevant as knowledge evolves, and often fail to capture model capabilities in specialized domains (Kiela et al., 2021; Dominguez-Olmedo et al., 2024; Zhang et al., 2024; Zhu et al., 2023; Ruder, 2023). While direct human assessment provides valuable insights, its cost and scalability limitations render it impractical for the continuous, diverse evaluation needs of the field. This creates a pressing need for evaluation generation frameworks that are automatic, while dynamic, reliable, domain-specific, and accessible.

We therefore introduce **YourBench**: an open-source framework that enables automated generation of bespoke evaluation sets directly from any collection of documents. YourBench empowers users to systematically create fresh, relevant benchmarks tailored to specific topics, achieving high reliability at low cost and without manual annotation. Central to our framework is the principle of Document-to-Evaluation Generation (D2EG), where LLMs are leveraged to produce diverse, contextually-grounded question-answer pairs with verifiable citations, optimizing for coverage, diversity, and answerability (details in §2.2, Appendix C).

We demonstrate the power of this approach by replicating the challenging MMLU-Pro benchmark (Wang et al., 2024). Using only a handful of Wikipedia articles as input, YourBench automatically generated novel, multiple-choice questions across 11 domains. When evaluated on 86 models, from 400M to 405B parameters, the resulting model rankings achieve

---
*Equal contribution.

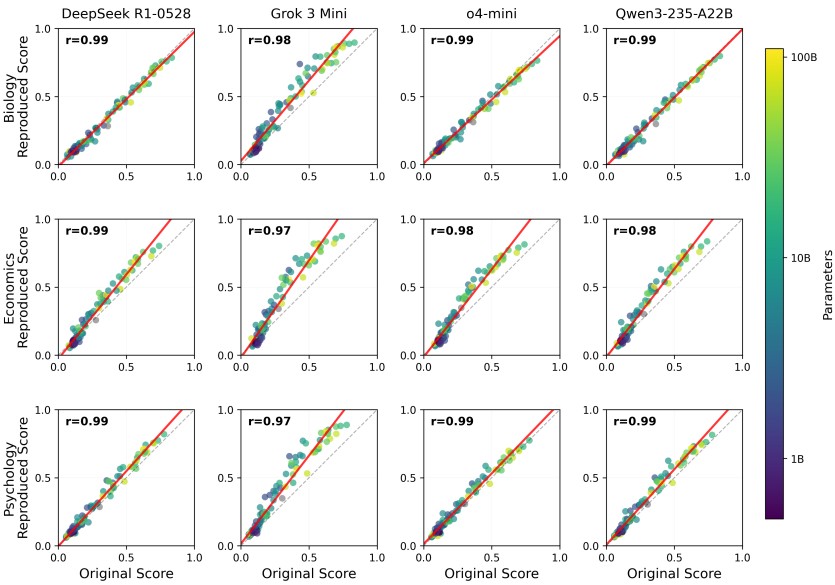

Model Performance Correlation: Original vs Reproduced Tasks

Figure 1: **YourBench Reliably Reproduces Expert-Curated Benchmarks.** We automatically reproduced MMLU-Pro across 11 domains using only Wikipedia articles. The scatter plots show the strong correlation between model scores on the original MMLU-Pro (x-axis) and our generated benchmark (y-axis) for Biology, Economics, and Psychology, each tested with four different generator models. Each point represents one of 86 evaluated models, demonstrating high fidelity (Pearson $\rho > 0.91$) across the board.

a remarkable 0.91 - 0.99 Pearson correlation with the original expert-curated benchmark (Figure 1). This process is not only reliable but also highly efficient, generating a fresh, contamination-proof evaluation for under \$15 per model.

The YourBench framework is built on a robust pipeline that includes multi-format document ingestion, semantic chunking, and automated quality controls like citation grounding (§2). To ensure our method evaluates true contextual reasoning rather than memorized knowledge, we also introduce TEMPORA-0325, a dataset of over 7,000 documents published after March 2025, providing a sandbox for evaluating models on genuinely novel information.

Our main contributions are as follows:

- **YourBench:** An open-source framework[1] for the automated generation of reliable, domain-specific evaluation benchmarks from any document set.

- **TEMPORA-0325:** A large-scale dataset[2] of recent, post-cutoff documents designed to enable robust, contamination-free evaluation.

- **State-of-the-Art Benchmark Replication:** A large-scale validation showing YourBench reproduces MMLU-Pro rankings across 86 models with near-perfect correlation, demonstrating the viability of automated benchmark generation.

By providing a scalable, automated, and document-grounded approach, YourBench facilitates a move towards more timely, specific, and trustworthy LLM evaluation. It empowers the community to better understand and track the true capabilities of these rapidly advancing models, ensuring that evaluation can finally keep pace with innovation.

---

[1]GitHub
[2]Dataset

# 2 YourBench: Multistep Framework for Dynamic Evaluation Generation

The YourBench framework is architected as a multi-stage pipeline designed to transform unstructured documents into high-fidelity evaluation benchmarks. Each stage addresses a specific challenge in automated generation, moving from raw content ingestion to a polished, reliable set of questions. This section details the three core stages: (1) Document Preprocessing, (2) Question and Answer Generation, and (3) Quality Filtering.

## 2.1 Document Preprocessing

Real-world documents are messy, arriving in diverse formats (PDF, HTML, Word) with varied structures and multimodal content. To create a consistent foundation for analysis, YourBench first employs a robust preprocessing pipeline. This pipeline standardizes all inputs into a clean, LLM-friendly markdown format while preserving critical semantic information. The process involves three key steps:

1. **Document Ingestion:** We normalize heterogeneous source files into a unified markdown representation. This step handles format conversion and extracts textual descriptions from visual elements, making all content accessible.

2. **Semantic Chunking:** To manage the context limitations of LLMs and focus their attention, we partition documents into semantically coherent segments. This avoids splitting key ideas and enables the generation of questions grounded in localized context.

3. **Global Summarization:** Chunking risks losing the document's broader narrative. We mitigate this by generating a global summary for each document, which provides essential overarching context to the model during question generation.

The detailed methodologies for this stage, including specific models and tools used for ingestion and chunking strategies like multi-hop combination, are elaborated in Appendix B.

## 2.2 Question and Answer Generation to Elicit Grounded Knowledge

The core of our framework is the *Document-to-Evaluation Generation* (D2EG) process, which aims to produce a question set that satisfies three principles: **Coverage** of the source material, **Diversity** in question style and difficulty, and **Answerability** directly from the provided text. To achieve this, YourBench employs a guided, ensemble-based approach.

We frame the generation task as sampling a question-answer pair $(q, a)$ and its supporting citations cit from a model conditioned on both the global document summary $S$ and a specific local chunk $c$:

$$(q, a, \text{cit}) \sim p(\cdot | \text{prompt}_{\text{gen}}, S, c) \tag{1}$$

Providing both the local chunk $c$ and the global summary $S$ is crucial; the local context provides the specific details for grounding, while the global summary ensures the model correctly interprets those details within the document's full scope (Liu et al., 2023). To ensure answerability, the generation prompt explicitly instructs the model to provide citations that substantiate its answer.

To enhance robustness and diversity, we utilize an ensemble of LLMs from different model families and scales ($\mathcal{M} = \{M_1, ..., M_N\}$). By aggregating outputs from multiple models, we mitigate the inherent biases and stylistic tendencies of any single model, resulting in a more comprehensive and varied raw question pool.

## 2.3 Quality Filtering and Deduplication: Ensuring Fidelity

The raw output from the LLM ensemble requires rigorous filtering to ensure fidelity and non-redundancy. YourBench implements a two-stage automated process: citation validation followed by semantic deduplication.

### 2.3.1 Citation Validation

To enforce that QA pairs are verifiably grounded in the source text, we first validate the model-provided citations. We implement an algorithmic check that quantifies the correspondence between each provided citation string and the source chunk using a fuzzy string matching score (based on Levenshtein distance (Levenshtein, 1966)). This produces a grounding score for each QA pair, and we filter out any pairs that fall below a configurable threshold $\theta_{cit}$, programmatically removing ungrounded or hallucinated content.

We assign a grounding score to each QA pair $(q, a, cit)$ by averaging the partial ratios across its citations:

$$\text{Score}_{QA}(q, a, cit) = \frac{1}{N_c} \sum_{i=1}^{N_c} \text{PartialRatio}(c_i, c) \tag{2}$$

assuming $N_c > 0$ (score is 0 if $N_c = 0$). We filter $Q_{raw}$, retaining pairs exceeding a threshold $\theta_{cit}$:

$$Q_{cit} = \{(q, a, cit) \in Q_{raw} \mid \text{Score}_{QA}(q, a, cit) > \theta_{cit}\} \tag{3}$$

Empirically, $\theta_{cit} = 0.85$ balances rigorous filtering of ungrounded pairs with preservation of valid items. See Appendix D.1 for the model-level scoring metric used in evaluations.

### 2.3.2 Semantic Deduplication and Reweighting

Ensemble generation and chunk overlap can lead to semantic redundancy in $Q_{cit}$. To manage this, we perform semantic deduplication. We obtain dense embeddings $e(q)$ for questions in $Q_{cit}$ using a sentence embedding model (e.g., Sentence-BERT (Reimers & Gurevych, 2019)).

We apply DBSCAN (Ester et al., 1996), a density-based clustering algorithm, to the embeddings $\{e(q)\}$. DBSCAN groups semantically similar QA pairs (cosine similarity $> \tau_{sim} = 0.9$) into clusters $\mathcal{C} = \{C_1, ..., C_K\}$ and identifies outliers $N$.

From each cluster $C_k$, we select one representative QA pair $(q_k^*, a_k^*, cit_k^*)$ (e.g., the medoid). The deduplicated set is:

$$Q_{dedup} = \{(q_k^*, a_k^*, cit_k^*) \mid C_k \in \mathcal{C}\} \cup N' \tag{4}$$

where $N'$ are the unique noise points. To retain information about concept salience (indicated by cluster size $|C_k|$), we assign weights $w_k$ to each representative $(q_k^*, a_k^*, cit_k^*)$ proportional to its original cluster size (e.g., $w_k = |C_k|$), with $w = 1$ for noise points. These weights are used in the final evaluation scoring (Section 3), allowing frequently questioned concepts to contribute more significantly, approximating the evaluation of the full set $Q_{cit}$ efficiently.

## 2.4 Suggested Evaluator

Given the curated, weighted QA set $Q_{final} = Q_{dedup}$ (Sections 2.2, 2.3), we generally evaluate free form LLMs outputs using a pairwise comparative assessment strategy (as is done in model arenas). Our suggested evaluator is composed of a judge LLMs ensemble to enhance reliability and mitigate self-preference bias (Zheng et al., 2023), and an bias-corrected scoring aggregation to mitigate positional bias (the tendency of LLMs-judges to prefer an answer presented in one position compared to the other). We expand on this in Appendix D.2. It's also possible to use YourBench to generate questions with multiple choice answers through prompt modifications, in which case it becomes possible to evaluate models through a simple exact match score, as we do in Section 3.3.

## 3 Empirical Validation and Results

We conduct a multi-faceted validation of YourBench to answer two central questions. First, can our automated framework produce benchmarks that reliably replicate the results of established, human-curated evaluations at scale? Second, are the questions generated by YourBench intrinsically high-quality, i.e, valid, diverse, and verifiably grounded in the source material?

## 3.1 Benchmark Replication: Matching Expert Curation at Scale

A critical test for any new evaluation framework is whether its results align with a trusted gold standard. To validate YourBench's external validity, we performed a large-scale replication of the challenging MMLU-Pro benchmark (Wang et al., 2024).

**Methodology.** For 11 MMLU-Pro domains (e.g., Biology, Economics, Physics), we used YourBench to automatically generate novel 10-choice multiple-choice questions. The source material was intentionally generic: the first 10 relevant Wikipedia articles for each domain. To test the robustness of our framework, we used four different generator models, including both open-source (DeepSeek R1 0528, Qwen3-235B) and closed-source (o4-mini, Grok-3-mini) options. We then evaluated a diverse set of 86 models, ranging from 400M to 405B parameters, on both the original MMLU-Pro and our generated benchmarks.

**Results.** YourBench demonstrates remarkably high-fidelity reproduction. As shown in Table 1, the Pearson correlation between model rankings on the original and our generated benchmarks is exceptionally high, ranging from **0.91 to 0.99** on average across domains. This strong correlation, visually confirmed by the tight clustering in Figure 1, holds true regardless of the generator model used, proving that our framework reliably preserves the relative capabilities of different LLMs. By aggregating scores across hundreds of generated questions, the evaluation effectively minimizes noise and captures the true signal of model performance. This entire process is also highly efficient, requiring under 5 minutes of computation and less than $15 in inference costs per domain. This result provides strong evidence that automated, dynamic benchmark generation can match the discriminative power of static, expert-curated evaluations.

Table 1: Pearson correlation between original MMLU-Pro and YourBench reproductions across 11 domains. Each correlation is computed over 86 models' performance scores, demonstrating consistently high fidelity regardless of the generator model used.

| Domain | YourBench Generator Model | | | |
|---|---|---|---|---|
| | DeepSeek R1 0528 | Grok-3-mini | o4-mini | Qwen3-235B |
| Biology | **0.994** | 0.977 | **0.994** | **0.994** |
| Business | 0.943 | 0.924 | 0.934 | 0.937 |
| Chemistry | 0.935 | 0.900 | 0.909 | 0.911 |
| Computer Science | 0.950 | 0.914 | 0.943 | 0.933 |
| Economics | 0.985 | 0.969 | 0.980 | 0.981 |
| Health | 0.971 | 0.949 | 0.966 | 0.968 |
| History | 0.933 | 0.898 | 0.952 | 0.943 |
| Law | 0.934 | 0.884 | 0.921 | 0.914 |
| Philosophy | 0.964 | 0.937 | 0.957 | 0.956 |
| Physics | 0.933 | 0.899 | 0.919 | 0.919 |
| Psychology | **0.991** | 0.975 | **0.990** | 0.989 |
| **Average** | **0.958** | 0.930 | 0.951 | 0.950 |

## 3.2 Internal Quality Validation: Are the Generated Questions Good?

Beyond replicating rankings, a trustworthy benchmark must consist of questions that are valid, diverse, and verifiably grounded. To analyze these intrinsic properties, we move from external replication to internal quality assessment.

**Experimental Setup.** To ensure this analysis measured true generative capabilities rather than memorization, we created the TEMPORA-0325 dataset, a corpus of over 7,000 diverse documents published exclusively after March 2025. This guarantees that all source material is novel to the generator models. We then used a suite of 26 state-of-the-art LLMs, spanning

7 model families and sizes from 3B to 671B parameters, to generate over 150,000 QA pairs from this dataset.

**The Validity-Diversity Spectrum.**   We analyzed the generated questions along two axes: validity and diversity. **Validity**: a question's clarity, sensibility, and answerability from the source text, was assessed via meticulous human evaluation of over 2,000 sampled questions, achieving high inter-annotator agreement (Gwet's AC1 = 0.71). **Diversity**: the semantic breadth of topics covered, was measured using embedding-based metrics.

Our analysis, summarized in Figure 2, reveals an intriguing trade-off. We found that contemporary models integrated within YourBench can generate questions with high intrinsic validity, averaging approximately 85% post-filtering. However, models exhibit distinct "personalities": some, like o3 mini, excel at generating highly valid questions (96%) by focusing on precise, factual queries, resulting in lower diversity. Others, like Qwen2.5 32B, achieve high diversity by exploring a wider range of concepts, at the cost of slightly lower average validity. Notably, some models like DeepSeek V3 achieve a strong balance, scoring well on both dimensions. This spectrum allows practitioners to select generator models that align with specific evaluation goals, whether for rigorous factual recall or broad conceptual understanding.

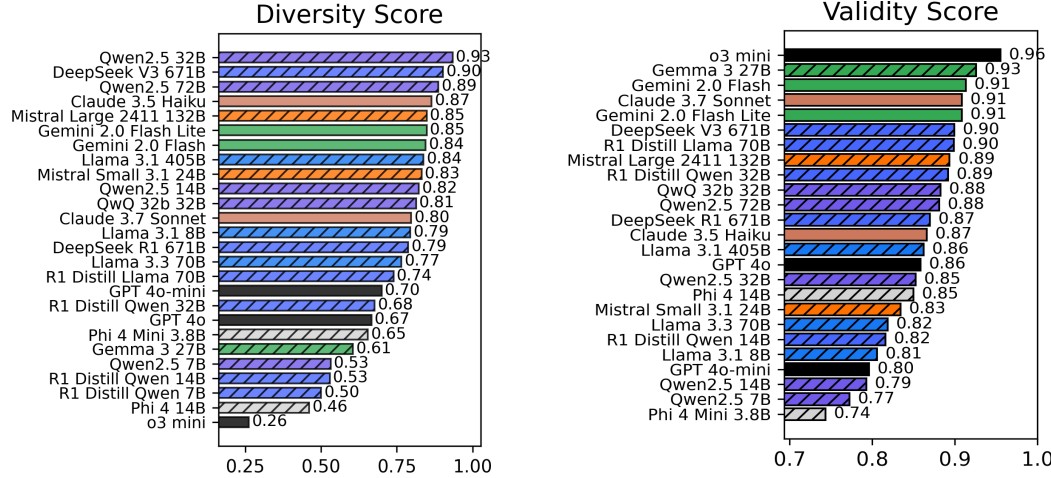

Figure 2: **The Validity-Diversity Spectrum of Language Models.** Semantic diversity scores (left) and human-annotated validity scores (right) for questions generated by various models. Models like o3 mini excel in validity but have low diversity, while models like Qwen2.5 32B achieve high diversity with slightly lower validity. Some models like DeepSeek V3 demonstrate a strong balance.

### 3.3 Citation Grounding: Ensuring Verifiable Fidelity

We enforce reliability through an automated citation validation step that scores the textual overlap between model-provided citations and the source text (§2.3). To assess model proficiency at this task, we computed an aggregate citation score for each generator.

Figure 3 shows that leading models, including both proprietary (Claude 3.7 Sonnet) and open-weight (Qwen, Gemma families) options, demonstrate strong citation capabilities. The Pareto frontier analysis in panel (b), which plots inference cost against citation score, further reveals that high-quality grounding is attainable efficiently. Models like Qwen2.5 32B achieve excellent citation scores at a fraction of the cost of top performers, demonstrating that reliable, verifiable benchmark generation is accessible even with modest resources.

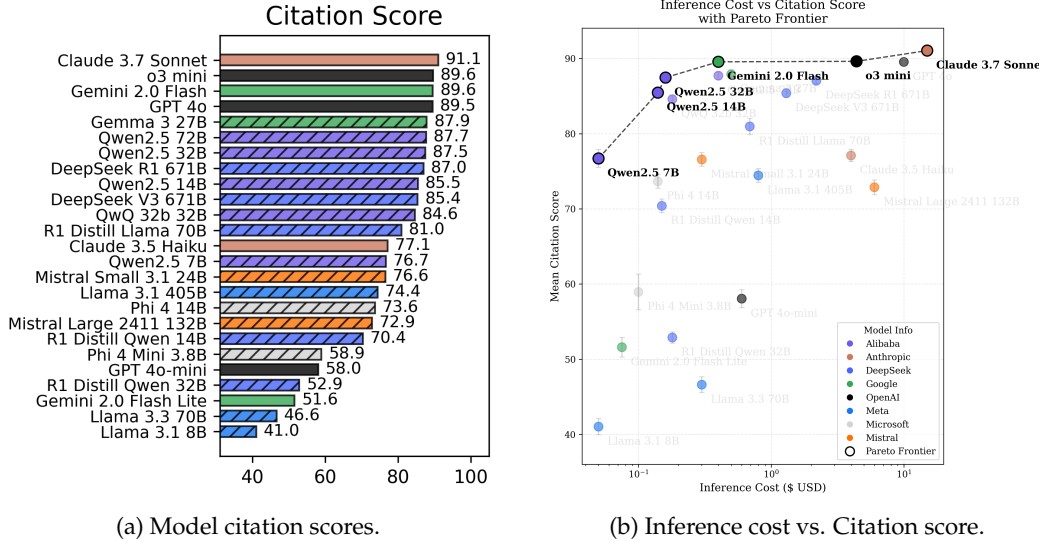

(a) Model citation scores.

(b) Inference cost vs. Citation score.

Figure 3: Evaluation of citation grounding performance. (a) Compares aggregate citation scores across various models. (b) Illustrates the Pareto frontier for inference cost (log scale) versus citation score, highlighting efficiency trade-offs. Full model list in Appendix D.3.

## 4    Related Work

The evaluation of LLMs is at a critical juncture, as the pace of model development has rendered many traditional methodologies inadequate. Our work is situated at the confluence of four major challenges in the literature: the limitations of static benchmarks, the rise of dynamic evaluation, the problem of temporal decay, and the need for domain-specific assessment.

**The Crisis of Static Benchmarks.**    Foundational benchmarks like GLUE (Wang et al., 2019) and MMLU (Hendrycks et al., 2021a) were instrumental in driving early progress. However, they now face two existential threats: **saturation**, where state-of-the-art models quickly reach or exceed human performance, diminishing the benchmark's discriminative power (Ruder, 2023); and **data contamination**, where test sets are inadvertently memorized during pretraining on vast web-scale corpora (Dominguez-Olmedo et al., 2024; Zhang et al., 2024). This forces a continuous and costly race to develop new static tests, which themselves are vulnerable to the same fate. YourBench directly addresses this cycle by enabling the instant generation of novel, contamination-free questions, making evaluation a dynamic rather than static target.

**The Move Towards Dynamic and Synthetic Evaluation.**    In response to these issues, the field has shifted towards dynamic evaluation. Early pioneering work like Dynabench (Kiela et al., 2021) introduced a human-in-the-loop adversarial process to find model weaknesses. While effective, this approach remains expensive and slow. More recent work has leveraged LLMs themselves to synthetically generate new evaluation data, either by rewriting existing questions or creating them from scratch (Wang et al., 2024). These methods, however, often struggle with quality control, risking the generation of trivial or hallucinated questions, and do not inherently solve the problem of grounding evaluation in specific, user-defined knowledge contexts. YourBench builds on this paradigm but introduces a structured, document-grounded pipeline with explicit quality controls like citation validation to ensure high fidelity.

**Temporal Validity and Domain Specificity.**    Two critical dimensions often unaddressed by general-purpose synthetic generation are time and domain. Knowledge evolves, and models pretrained on historical data quickly become outdated, performing poorly on questions

about recent events or concepts (Zhu et al., 2023). Simultaneously, high performance on general benchmarks often fails to translate to specialized, high-stakes domains like law or medicine, where precision and current knowledge are paramount (Holzenkamp et al., 2023; Hung et al., 2023b). Creating benchmarks for these areas has traditionally required expensive, slow expert annotation. YourBench provides a unified solution to both problems. By ingesting any set of documents, whether recent news articles (like our TEMPORA-0325 dataset) or proprietary domain-specific texts, it empowers any user to create timely, specialized evaluations in minutes, a task that was previously infeasible.

## 5  Conclusion

In this work, we introduced **YourBench**, an open-source framework that fundamentally changes how large language models are evaluated. By enabling the automated generation of reliable, document-grounded benchmarks, YourBench directly addresses the critical bottlenecks of static evaluation: contamination, saturation, and prohibitive cost. Our large-scale replication of MMLU-Pro, which achieved 0.91 - 0.99 Pearson correlation with the expert-curated original using only a handful of Wikipedia articles, provides definitive evidence that automated generation can match the discriminative power of human-curated benchmarks at a fraction of the cost.

The utility of YourBench extends beyond benchmark replication. By empowering researchers and practitioners to create custom evaluations from any document set, it unlocks new possibilities for domain-specific and temporally-aware assessment.

The framework's potential extends beyond benchmark replication and is already being explored in several research initiatives:

- **Domain-Specific Knowledge Assessment (Agriculture):** YourBench is being utilized to systematically evaluate LLMs on specialized, proprietary knowledge. This involves generating questions assessing factual recall, applied reasoning, and retrieval-augmented generation capabilities based on diverse agricultural documents, effectively measuring a model's domain intelligence

- **Personalized Education:** In educational research, the framework is being adapted to assist teachers and generate tailored assessment questions based on individual student learning profiles derived from textual inputs, exploring pathways towards automated, personalized learning tools.

- **Advanced RAG Training Data:** YourBench's capacity for multi-hop question generation is being employed to construct challenging training corpora for retrieval-augmented generation systems. By synthesizing complex questions requiring information integration across multiple document chunks and incorporating human feedback loops, this effort aims to push the boundaries of RAG model capabilities.

By releasing the YourBench library, the contamination-free TEMPORA-0325 dataset, and all supporting code and data, we provide the community with a practical, scalable, and robust tool. We believe that dynamic, user-driven evaluation is essential for ensuring that our assessment methodologies can finally keep pace with the rapid evolution of AI capabilities.

## Reproducibility

We are committed to ensuring the reproducibility of our research and facilitating further investigation by the community. To this end, we make several key resources publicly available. The complete source code for the **YourBench** framework is released under an open-source license and can be accessed at https://github.com/huggingface/yourbench. This repository includes the implementation of the document processing pipeline (Section 2.1), the question generation framework (Section 2.2), and associated evaluation scripts.

Furthermore, the TEMPORA-0325 dataset introduced in Section 3, comprising documents published after March 1, 2025, is available on the Hugging Face Hub at this datasets link.

Alongside the dataset, we provide the code used for document collection, preprocessing, semantic chunking (Section B.2), and subsequent analysis within the main framework repository.

To enable detailed verification of our experimental findings, we release the complete inference traces for critical experiments, including the MMLU Pro replication study (Section 3.1) and the citation validity analysis (Figure 3). These traces cover the diverse set of 26 large language models detailed in Section 3, spanning both open-weight models (e.g., Llama, Qwen, DeepSeek families) and closed-source API-based models (e.g., GPT, Claude, Gemini families). Our inclusion of both model types is a deliberate choice to enhance long-term reproducibility; by providing results for open models, we ensure that future researchers can replicate or extend our core findings even if commercial APIs become deprecated or change significantly over time. All code and experimental artifacts are designed to support transparency and allow the community to build upon our work effectively.

## Ethical Considerations

The development of powerful AI systems necessitates equally robust and trustworthy methods for their evaluation. Frameworks like YourBench, which automate the generation of evaluation benchmarks, represent a step towards more dynamic and potentially less contaminated assessment. However, like any technology, its introduction warrants careful consideration of the ethical dimensions and potential societal impacts.

One important area relates to the human element in data creation. Traditionally, benchmark creation involves significant human labor, often in the form of detailed annotation or question writing. This labor, while essential, can sometimes be repetitive and subject to economic pressures, including concerns about fair compensation, particularly in globally distributed workforces. YourBench introduces a potential shift in this dynamic. By automating the generation of question-answer pairs, the burden on humans might transition from primarily generative tasks to ones involving oversight, validation, and curation. Instead of authoring questions from scratch, the focus could shift towards assessing the quality, relevance, and safety of machine-generated content, or guiding the generation process towards specific evaluation goals. It's uncertain as of now whether such a shift would rather elevate the nature of the work, (demanding more critical judgment rather than repetitive production), or simply remove large-scale, low-wage annotators from the equation by replacing them with skilled annotators. It requires careful consideration and proactive effort to ensure that individuals involved are equipped with the necessary skills for these evolving roles and that the economic benefits of automation are shared equitably. The potential for deskilling or displacement in certain areas must also be acknowledged and addressed thoughtfully by the community and organizations deploying such systems. We must remain mindful of the human collaborators whose insights remain crucial, even as the tools evolve.

Furthermore, the integrity of the evaluation process itself relies heavily on the quality and characteristics of the LLMs used within the YourBench framework. The models employed for generating questions, summaries, and even judging responses inevitably embed their own biases, limitations, and potential failure modes, learned from their own training data. If not carefully managed, YourBench could inadvertently propagate or even amplify these biases within the generated benchmarks. This underscores the critical importance of transparency regarding the models used in the generation process and the need for robust, ongoing validation of the generated datasets – not just for correctness, but also for fairness, representation, and potential hidden biases. Automated checks, like the citation grounding implemented, are valuable, but **human oversight remains essential for identifying more subtle issues**.

The increased accessibility offered by YourBench, allowing for rapid generation of domain-specific benchmarks, is a significant advantage. It empowers researchers and practitioners to create evaluations tailored to their specific needs, moving beyond generic, potentially saturated benchmarks. However, this ease of creation also carries a potential for misuse. Benchmarks could conceivably be generated to specifically highlight the strengths or weak-

nesses of particular models, potentially leading to misleading comparisons if not used responsibly and transparently.

Finally, the computational resources required to run ensembles of large models for generation and evaluation contribute to the environmental footprint of AI development. While YourBench might offer efficiencies compared to certain manual processes or continuous large-scale human evaluations, the aggregate energy consumption remains a factor worthy of consideration as such automated systems become more widespread.

In conclusion, while YourBench offers a promising direction for advancing LLM evaluation, its development and deployment must proceed with a deep sense of responsibility. Continuous monitoring of its impacts, particularly on human labor dynamics and the integrity of evaluation results, is essential. The goal should not merely be automation, but the creation of evaluation methodologies that are not only more efficient and relevant but also fundamentally fair, trustworthy, and aligned with the broader goal of developing beneficial AI.

## Acknowledgements

This research project has benefited from the Microsoft Accelerate Foundation Models Research (AFMR) grant program through which leading foundation models hosted by Microsoft Azure, along with access to Azure credits, were provided to conduct the research. Additionally, this research utilized Anthropic credits granted through Anthropic's External Researcher Access Program. This research used the Delta advanced computing and data resource, supported by the National Science Foundation (award OAC 2005572) and the State of Illinois; Delta is a joint effort of the University of Illinois Urbana-Champaign and its National Center for Supercomputing Applications. We also gratefully acknowledge Hugging Face for supporting inference costs, as well as SambaNova and Novita for providing inference services.

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

## A  YourBench Pipeline Overview

Figure 4 provides a high-level schematic of the end-to-end YourBench framework. The process begins with ingesting diverse source documents, which are then preprocessed through steps like semantic chunking and summarization (§2.1, Appendix B). An ensemble of LLMs generates raw question-answer pairs grounded in the document chunks, guided by principles aiming for coverage, diversity, and answerability (§2.2, Appendix C). These raw outputs undergo rigorous quality filtering, including citation validation and semantic deduplication, to produce a high-fidelity evaluation set (§2.3). Finally, this curated set is used within an automated evaluation framework, typically employing an ensemble of LLM judges to rank the performance of target models (§3). This modular pipeline allows for flexibility and robust, automated benchmark creation from arbitrary document inputs.

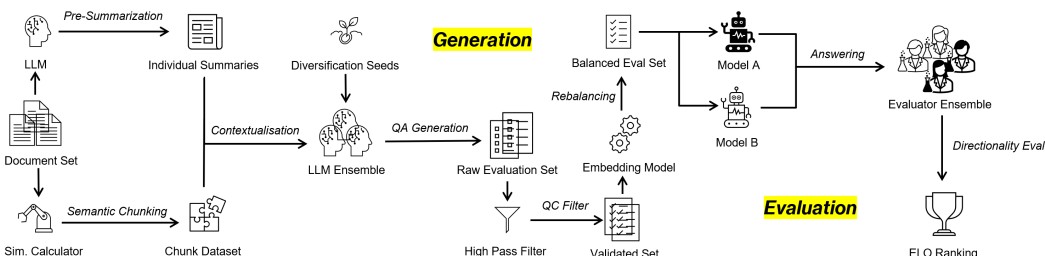

Figure 4: Overview of the YourBench Framework: A dynamic pipeline starting from diverse documents, through preprocessing (ingestion, chunking, summarization - §2.1), LLM-driven question generation (following D2EG principles - §2.2), quality filtering (citation validation, deduplication - §2.3), to automated evaluation using an LLM judge ensemble (§3).

## B  Detailed Document Preprocessing

This appendix details the multi-stage preprocessing pipeline used in YourBench, designed to convert diverse, real-world documents into a standardized format suitable for LLM-based question generation, as summarized in Section 2.1. The pipeline addresses challenges posed by heterogeneous formats and multimodal content.

### B.1  Document Ingestion

We implement a unified ingestion pipeline using ReaderLM-v2 (Wang et al., 2025) (stripping known HTML content) and Markitdown[3] (converting various document types like PDF and Word into markdown). This approach retains key structural elements (headings, lists, tables, math) while simplifying complex layouts into a standard text-based markdown syntax, ensuring consistency across sources.

When visual content (e.g., images) is present, we generate high-level descriptions using Qwen2.5-72B-VL (Team, 2025) for captioning. These descriptions are incorporated into the markdown representation, allowing question generation modules to reference both textual and visual information uniformly. An example of a multimodal document input is shown in Appendix B.4 (Figure 5).

### B.2  Semantic Chunking

Processing full documents directly with LLMs presents challenges, including attention dispersion potentially overlooking content (Ye et al., 2024), and performance degradation with longer contexts (Liu et al., 2023).

---

[3]https://github.com/microsoft/markitdown

We address these through semantic chunking, which partitions documents into coherent segments. This process involves decomposing the document into sentences, computing embeddings, and then splitting the text into chunks based on semantic similarity and token length constraints, preserving coherence within and across segments. Multi-hop chunking is also implemented by combining multiple non-contiguous chunks to facilitate questions requiring information synthesis across different document parts.

Given a document $d$, we first decompose it into sentences $S = \{s_1, ..., s_n\}$ and compute their embeddings $E = \{e_1, ..., e_n\}$ using a sentence transformer model (Reimers & Gurevych, 2019), where $e_i \in \mathbb{R}^k$. The chunking process is governed by three parameters: $l_{min}$: minimum chunk length in tokens, $l_{max}$: maximum chunk length in tokens, and $\tau$: similarity threshold for chunk boundaries. For consecutive sentences $s_i$ and $s_{i+1}$, we compute their semantic similarity using cosine similarity:

$$sim(s_i, s_{i+1}) = \frac{e_i \cdot e_{i+1}}{\|e_i\| \|e_{i+1}\|}$$

A chunk **boundary** is established at position $i$ when the current chunk's token length exceeds $l_{min}$ AND either $sim(s_i, s_{i+1}) < \tau$ OR appending $s_{i+1}$ would cause the accumulated chunk length to exceed $l_{max}$. This process yields a set of text chunks $C = \{c_1, ..., c_m\}$ where each chunk $c_j$ is a contiguous sequence of sentences from $S$.

**Multihop Chunking:** To enable the generation of questions requiring synthesis across multiple document segments, we implement multihop chunking. Given parameters $h_{min}$ and $h_{max}$ (minimum and maximum number of hops), we generate composite chunks. For each multihop chunk, we sample $k \sim \mathcal{U}(h_{min}, h_{max})$ original chunks uniformly without replacement from $C$ and concatenate their text content. This produces a set of multihop chunks $M = \{m_1, ..., m_p\}$ where each $m_i$ consists of $k$ potentially non-contiguous original chunks. These multihop chunks are used alongside the original chunks $C$ during question generation (Section 2.2). appendix context

### B.3 Document Summarization

While chunking manages context length, it can lead to a loss of global document perspective during question generation. To mitigate this, we generate a document-wide summary using an LLM (DeepSeek-V3 (DeepSeek-AI et al., 2025b) with zero temperature). For extremely long documents exceeding context limits, techniques like those in (Chang et al., 2024) can be employed. Our summarization uses chain-of-thought prompting (Wei et al., 2023) with structured XML tags[4] for quality and consistency. This concise summary is provided alongside individual chunks (Section 2.2) to give the question generation LLM both local detail and global context. The full summarization prompt is available in Appendix H.

### B.4 Sample Document

Figure 5 shows an example document typical of those included in the dataset, featuring a mix of text and visual elements handled by our preprocessing pipeline (Appendix B).

---

[4]https://docs.anthropic.com/en/docs/build-with-claude/prompt-engineering/use-xml-tags

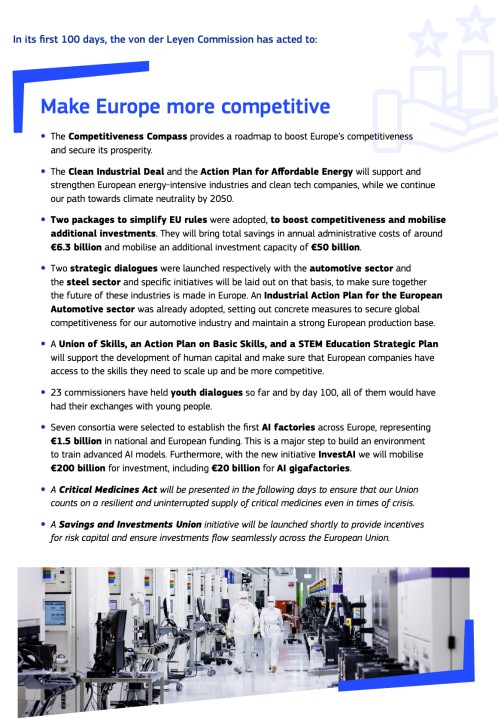

Figure 5: Example of a contemporary multimodal document included in Tempora-0325

## C  Theoretical framework underlying the data generation work

This appendix outlines the theoretical foundation for automated benchmark generation from source documents within the YourBench framework, termed *Document-to-Evaluation Generation* (D2EG), as introduced conceptually in Section 2.2. The goal is to produce a representative question set $Q$ derived from a source document (partitioned into segments $\{c_1, \ldots, c_m\}$ and optionally summarized by $s$) that satisfies key desiderata.

Let $\mathcal{Q}$ be the universe of all possible questions derivable from the document segments. We seek a subset $Q \subseteq \mathcal{Q}$ that optimizes the trade-off between:

1.  **Minimality:** Penalizing the total number of questions $|Q|$ for efficiency.
2.  **Coverage:** Maximizing the extent to which $Q$ addresses the source material.
3.  **Diversity:** Ensuring variety in question type, difficulty, and targeted reasoning skills.
4.  **Answerability & Quality:** A hard constraint ensuring every $q \in Q$ is valid and verifiably answerable from the source.

This can be formalized conceptually as a constrained optimization problem:

$$\min_{Q \subseteq \mathcal{Q}} \mathcal{L}(Q) = \alpha \, |Q| + \beta \, \mathcal{L}_{\text{uncov}}(Q) + \gamma \, \mathcal{L}_{\text{unif}}(Q), \tag{5}$$

subject to the constraint that every question in $Q$ is verifiably answerable from the source document. Here, $\mathcal{L}_{\text{uncov}}(Q)$ penalizes the amount of source material left uncovered by $Q$, while $\mathcal{L}_{\text{unif}}(Q)$ penalizes lack of diversity (e.g., high semantic overlap) within $Q$. The non-negative coefficients $\alpha, \beta, \gamma$ balance these competing objectives.

Finding an exact solution to (5) is generally intractable due to the combinatorial nature of selecting $Q$ from $\mathcal{Q}$. Therefore, as described in Section 2.2, YourBench employs a practical, greedy generation framework using LLMs, guided by prompts and context, to approximate a desirable question set $Q$ that adheres to the D2EG principles.

# D  Framework Theoretical Complements

## D.1  Citation validity

To validate the grounding of a generated answer $a$ with citations $\text{cit} = \{c_1, ..., c_{N_c}\}$ to a source text chunk $c$, we use fuzzy string matching. For a given citation string $c_i$ and the source chunk text $c$, we compute the partial ratio score using the Levenshtein distance concept:

$$\text{PartialRatio}(c_i, c) = \max_{s_j \subseteq c} \frac{2 \cdot \text{LCS}(c_i, s_j)}{|c_i| + |s_j|} \times 100$$

where $\text{LCS}(c_i, s_j)$ is the length of the longest common subsequence between the citation $c_i$ and a substring $s_j$ of the source text $c$. The maximum is taken over all possible substrings $s_j$ of $c$. This score ranges from 0 to 100.

The overall grounding score for a single QA pair $(q, a, \text{cit})$ is calculated as described in Section 2.3 (Eq. (2)).

To calculate an overall citation performance score for a specific *generation model* (as reported in Section D.1), we average the QA grounding scores across all questions generated by that model:

$$\text{ModelCitationScore} = \frac{1}{N_{q,\text{model}}} \sum_{q=1}^{N_{q,\text{model}}} \text{Score}_{\text{QA}}(q, a_q, \text{cit}_q)$$

where $N_{q,\text{model}}$ is the total number of valid questions generated by the model after initial filtering, and $\text{Score}_{\text{QA}}(q, a_q, \text{cit}_q)$ is the grounding score for question $q$ as defined in Eq. (2).

## D.2  Evaluation Framework

Given the curated, weighted QA set $Q_{\text{final}} = Q_{\text{dedup}}$ (Sections 2.2, 2.3), we evaluate LLMs $\mathcal{M} = \{M_1, ..., M_N\}$ using a pairwise comparative assessment strategy with an ensemble of judge LLMs $\mathcal{J} = \{J_1, ..., J_K\}$ to enhance reliability (Zheng et al., 2023).

For each question $(q_j, a_j^*, \text{cit}_j^*) \in Q_{\text{final}}$ (weight $w_j$) and model pair $(M_A, M_B)$, we elicit responses $R_A^j, R_B^j$. Each judge $J_l \in \mathcal{J}$ receives the context tuple:

$$\xi_{j,l,A,B} = (q_j, R_A^j, R_B^j, S, c_j) \tag{6}$$

including the question $q_j$, responses $R_A^j, R_B^j$, global summary $S$, and source chunk(s) $c_j$ for grounded evaluation.

The judge $J_l$ produces a continuous score $v_{lj}(A, B) \in [-1, 1]$ reflecting the relative quality of $R_A^j$ vs $R_B^j$, often guided by a prompted chain-of-thought process (see Appendix for prompt details):

$$v_{lj}(A, B) = J_l(\xi_{j,l,A,B}) \tag{7}$$

Scores are averaged across judges for consensus $\bar{v}_j(A, B) = \frac{1}{K} \sum_{l=1}^{K} v_{lj}(A, B)$ and weighted by question salience $w_j$:

$$V_j(A, B) = w_j \cdot \bar{v}_j(A, B) \tag{8}$$

To counteract positional bias, we evaluate both $(A, B)$ and $(B, A)$ pairings and compute a bias-corrected score:

$$V_j'(A, B) = \frac{1}{2} \left( V_j(A, B) - V_j(B, A) \right) \tag{9}$$

The overall comparative score $S(A, B)$ between $M_A$ and $M_B$ is the sum over all questions:

$$S(A, B) = \sum_{j=1}^{|Q_{\text{final}}|} V_j'(A, B) \tag{10}$$

The sign indicates preference; magnitude indicates difference strength. These pairwise scores $\{S(A, B)\}$ form the basis for global ranking using methods like Bradley-Terry (Bradley & Terry, 1952) or Elo (Elo, 1978).

## D.3 Evaluated Models

The following 26 models from 7 families were used in the generation and evaluation experiments described in Section 3:

- **DeepSeek** (DeepSeek-AI et al., 2025b;a): DeepSeek V3 (671B), DeepSeek R1 (671B), DeepSeek R1-Distill-Llama (70B), and DeepSeek R1-Distill-Qwen (32B, 14B, 7B).

- **Qwen** (Qwen et al., 2025): Qwen2.5 models at various scales (72B, 32B, 14B, 7B) and the reasoning model Qwen QwQ (32B).

- **Mistral** (Jiang et al., 2023): Mistral Large 2411 (132B) and Mistral 3.1 Small (24B).

- **Llama** (Dubey et al., 2024): Llama 3.1 (405B, 8B) and Llama 3.3 (70B).

- **Google** (Team et al., 2024): Gemini 2.0 Flash, Gemini 2.0 Flash Lite (?B) and Gemma 3 (27B).

- **OpenAI** (OpenAI et al., 2024): GPT-4o, GPT-4o mini, and o3 mini (?B).

- **Anthropic** (Anthropic, 2024): Claude 3.7 Sonnet, Claude 3.5 Haiku (?B).

# E Evaluation Quality Details

This appendix provides detailed methodologies and supplementary results for the validation of generated evaluation quality presented in Section 3.2.

## E.1 Question Validity Methodology and Detailed Results

**Human Evaluation Setup.** As introduced in Section 3.2, we conducted a manual evaluation to assess the intrinsic quality of generated questions. We sampled 2,000 unique questions generated from the TEMPORA-0325B dataset (Section 3) using the models listed in Appendix D.3. The sampling was stratified to ensure representation across models, document domains, targeted difficulty levels (basic, advanced), and question types (e.g., factual, multi-hop, numeric) specified during generation (Section 2.2).

Twenty trained annotators participated. Each annotator was presented with the source document chunk(s), the global document summary, the generated question, and the model-generated answer with its citations. Annotators were asked to assign a binary validity label (Valid/Invalid) based on the following criteria:

- **Clarity:** Is the question grammatically correct and unambiguous?

- **Contextual Answerability:** Can the question be definitively answered using *only* the provided document chunk(s) and summary? Does it require external knowledge or unwarranted assumptions?

- **Sensibility:** Is the question reasonable and logically coherent in the context of the document? (e.g., not nonsensical or self-contradictory).

A question was marked "Valid" only if it met all three criteria positively. Any ambiguity, reliance on external knowledge, or nonsensical phrasing resulted in an "Invalid" rating.

**Inter-Annotator Agreement.** Each question was evaluated independently by 3 randomly assigned annotators. To measure the consistency of their judgments, we calculated Gwet's AC1 coefficient (Gwet, 2008), a robust statistic for assessing inter-rater reliability, especially suitable for binary ratings with potential prevalence issues. The formula for Gwet's AC1 for two raters is:

$$AC1 = \frac{P_a - P_e(\gamma)}{1 - P_e(\gamma)}$$

where $P_a$ is the observed percent agreement, and $P_e(\gamma)$ is the chance agreement probability, calculated as $P_e(\gamma) = 2\pi(1 - \pi)$, with $\pi$ being the overall proportion of "Valid" ratings (averaged across raters). For multiple raters (3 in our case), we used a multi-rater extension of the formula. The resulting overall AC1 score was 0.71, typically interpreted as substantial agreement (Landis & Koch, 1977), confirming the reliability of our human validity labels.

**Detailed Results and Examples.** The average validity rate reported in the main text (≈85%) represents the mean percentage of questions rated "Valid" (by majority vote across the 3 annotators) across all models and question types post-filtering. The per-model validity scores are visualized in Figure 2 (right panel). Further breakdowns (e.g., validity per question type) can be derived from the released annotations accompanying our dataset. Examples of questions marked "Valid" and "Invalid" during this process, illustrating common failure modes like ambiguity or requiring external knowledge, are provided in Appendix I.

Juxtaposing these results highlights a prevalent, though not absolute, trade-off. The model achieving the highest validity, o3 mini, scores lowest in diversity (0.26). This suggests a generative posture focused on precision and safety, perhaps by asking more routine or algorithmically verifiable questions based directly on easily identifiable facts, leading to high validity but low exploration of the document's semantic space. Conversely, the top diversity model, Qwen2.5 32B, while still generating reasonably valid questions (0.81 validity, rank #11), sacrifices some peak validity in favor of broader conceptual coverage. This might indicate a more exploratory or creative generation strategy.

This validity-diversity spectrum is not a strict dichotomy. Notably, models like DeepSeek V3 671B manage to achieve impressive scores on both metrics (0.90 diversity, rank #2; 0.90 validity, rank #6), suggesting that balancing breadth and correctness is achievable. Similarly, models like Claude 3.7 Sonnet (0.80 diversity, 0.91 validity) also perform well across both dimensions.

This observed tension between generating highly valid, focused questions versus diverse, exploratory questions is an intriguing phenomenon. It reflects the different latent capabilities and perhaps inherent strategies employed by various LLMs when tasked with abstracting knowledge into evaluative queries. Rather than a limitation, this presents a valuable characteristic of the YourBench framework: it allows practitioners to select generator models or ensembles that align with their specific evaluation goals—be it rigorous testing of factual recall with high-validity generators, or broad assessment of understanding across topics using high-diversity generators, or seeking a balanced perspective with models adept at both. Understanding this trade-off provides deeper insight into the nature of LLM-driven generation and empowers more informed benchmark creation.

**Length Metrics vs. Validity.** We also analyzed the relationship between question/answer/citation length and the observed validity rate from human evaluation. Figure 6 plots the validity rate (averaged across all models) against different length metrics binned appropriately. While there isn't a perfectly monotonic trend, we observe a general tendency for validity to decrease slightly for very long questions, answers, or unified text lengths, potentially reflecting the increased difficulty in maintaining coherence and contextual grounding over longer generations. Citation length shows less variation. The black line represents the average validity rate across bins, while faint lines show individual model trends, highlighting variability. These plots reinforce the finding that generating complex (often longer) valid questions remains challenging.

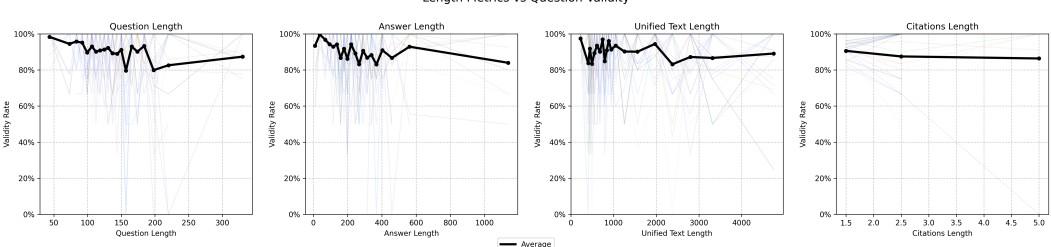

Figure 6: Relationship between generation length metrics and average question validity rate (across all models). Validity tends to decrease slightly for very long generations. Faint lines represent individual model trends.

### E.2 Citation Grounding Methodology and Detailed Results

**Citation Scoring Metric.** As described in Section 2.3, we quantify the grounding of an answer $a$ with citations $\text{cit} = \{c_1, ..., c_{N_c}\}$ to a source chunk $c$ using fuzzy string matching. The core metric is 'PartialRatio', based on Levenshtein distance (Levenshtein, 1966), computed for each citation $c_i$ against the source $c$:

$$\text{PartialRatio}(c_i, c) = \max_{s_j \subseteq c, |s_j| \geq |c_i|} \frac{2 \cdot \text{Match}(c_i, s_j)}{|c_i| + |s_j|} \times 100$$

where $\text{Match}(c_i, s_j)$ finds the length of the best matching contiguous block between $c_i$ and substrings $s_j$ of $c$ (typically using sequence matching algorithms). The maximum is taken over substrings $s_j$ of $c$ that are at least as long as the citation $c_i$. This score ranges from 0 (no match) to 100 (perfect match of $c_i$ within $c$).

The QA grounding score $\text{Score}_{\text{QA}}(q, a, \text{cit})$ is the average of these partial ratios across all $N_c$ citations, as given in Eq. (2). If $N_c = 0$, the score is 0.

**Model-Level Citation Score.** The overall citation score for a generation model $M$, as reported in Figure 3, is the average of the QA grounding scores across all valid QA pairs generated by that model:

$$\text{ModelCitationScore}_M = \frac{1}{|Q_{\text{valid},M}|} \sum_{(q,a,\text{cit}) \in Q_{\text{valid},M}} \text{Score}_{\text{QA}}(q, a, \text{cit})$$

where $Q_{\text{valid},M}$ is the set of QA pairs generated by model $M$ that passed initial quality filters (e.g., parseable format, non-empty question/answer). This provides a single metric to compare the average citation reliability of different models. Detailed scores for all evaluated models are implicitly represented in Figure 3.

**Inference Cost Calculation.** The inference costs used in Figure 3b were estimated based on the per-token pricing for output tokens (as generation is output-heavy) published on OpenRouter (https://openrouter.ai/docs/models) as of the time of experiments, using the lowest available price tier for each model. For models not on OpenRouter or without public pricing (indicated by "?B" parameters), relative cost estimates were made based on known parameter counts or comparable models where possible, or they were excluded from the cost analysis. This provides a practical estimate of the economic efficiency of using different models for generation within the YourBench framework.

### E.3 Semantic Diversity Methodology and Detailed Results

**Diversity Metrics.** As discussed in Section 3.2, we quantified the semantic diversity of the set of questions $Q_M$ generated by a model $M$ using two embedding-based metrics:

1. **Embedding Dispersion:** We first compute sentence embeddings $e(q)$ for each question $q \in Q_M$ using a standard sentence transformer model (e.g., 'all-mpnet-base-v2' (Reimers & Gurevych, 2019)). The dispersion is the average pairwise cosine distance:

$$\text{Dispersion}(Q_M) = \frac{1}{|Q_M|(|Q_M|-1)} \sum_{q_i \in Q_M} \sum_{q_j \in Q_M, i \neq j} \left( 1 - \frac{e(q_i) \cdot e(q_j)}{\|e(q_i)\| \|e(q_j)\|} \right)$$

A higher dispersion value indicates that the question embeddings are, on average, further apart in the embedding space, suggesting greater semantic variety.

2. **Semantic Entropy:** We apply K-Means clustering (with $K$ chosen based on heuristics like the elbow method or a fixed moderate number, e.g., $K = 50$) to the question embeddings $\{e(q) \mid q \in Q_M\}$. Let $N_k$ be the number of questions assigned to cluster $k$, and $N = |Q_M| = \sum_k N_k$. The proportion of questions in cluster $k$ is $p_k = N_k/N$. The semantic entropy is the Shannon entropy of the cluster distribution:

$$\text{Entropy}(Q_M) = - \sum_{k=1}^{K} p_k \log_2(p_k)$$

Higher entropy indicates that the questions are distributed more evenly across different semantic clusters, implying broader coverage of different conceptual areas. Lower entropy suggests concentration in a few dominant semantic themes.

The final "Diversity Score" reported in Figure 2 (left panel) is a normalized combination or average of these two metrics (e.g., scaled to [0, 1] based on observed ranges across models). This composite score aims to capture both the spread and the evenness of the semantic distribution.

**Detailed Scores.**   Figure 2 provides the final composite diversity scores for the evaluated models. The underlying dispersion and entropy values, along with the specific normalization method, are available with the project's source code and results data. The variation observed confirms that model choice significantly impacts the semantic breadth of the generated evaluation set.

### E.4   Cost and Parameter Efficiency Analysis

Beyond citation grounding (Figure 3b), we analyzed the relationship between model cost/size and overall question quality, approximated by the average validity score (Section 3.3). Figures 7a and 7b show Pareto frontiers for average validity score versus inference cost and model parameters, respectively.

These plots further illustrate favorable scaling trends and efficiency possibilities.

- **Cost Efficiency (Fig. 7a):** Models like Llama 3.1 8B, Gemini 2.0 Flash Lite, and Gemma 3 27B appear on or near the Pareto frontier, achieving relatively high validity scores (80-90%+) at substantially lower costs compared to the largest or most expensive models. This demonstrates that high question validity is attainable without exorbitant inference budgets.

- **Parameter Efficiency (Fig. 7b):** Smaller models, including Phi 4 Mini 3.8B, Qwen2.5 7B, Llama 3.1 8B, and Phi 4 14B, form part of the Pareto frontier. This indicates that smaller parameter counts do not necessarily preclude high validity generation. Phi 4 14B, for instance, reaches approximately 85% validity, competitive with much larger models, showcasing significant parameter efficiency. Gemma 3 27B also stands out, achieving over 90

Together, these analyses suggest that while larger models sometimes offer peak performance, carefully selected smaller or more cost-effective models can generate high-quality evaluation sets efficiently within the YourBench framework, democratizing access to customized benchmarking.

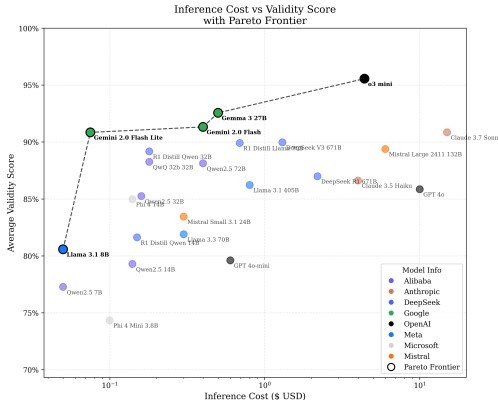
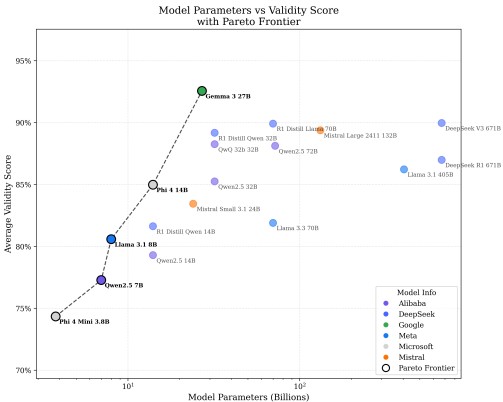

(a) Inference Cost vs. Average Validity Score.     (b) Model Parameters vs. Average Validity Score.

Figure 7: Pareto frontiers illustrating trade-offs between average question validity and (a) inference cost (log scale) and (b) model parameters (log scale). Smaller/cheaper models like Llama 3.1 8B, Gemini 2.0 Flash Lite, and Phi 4 14B can achieve high validity scores efficiently. Full model list in Appendix D.3.

# F MMLU Replication: Detailed Analysis and Results

Prior to our comprehensive MMLU-Pro validation (Section 3.1), we conducted preliminary experiments reproducing 7 subsets of the original MMLU benchmark with 8 models. While these initial results showed perfect Spearman correlation ($\rho = 1.0$) at the aggregate level, they were much more limited in number of evaluated models and domains, as well as overall number of questions.

This appendix provides a detailed breakdown of the MMLU replication experiment discussed in §3.1 and introduced in Figure 1. We aimed to validate whether YourBench could automatically generate MMLU-style benchmarks from source documents that reliably reflect the relative performance of different LLMs compared to the original MMLU benchmark.

## F.1 Correlation Analysis

We evaluated a suite of 8 LLMs (see Table 2) on 7 original MMLU subject subsets and their corresponding YourBench-generated counterparts ("new"). We then analyzed the correlation between the performance scores (accuracy) obtained on the original versus the "new" benchmarks.

- **Overall Correlation (All Subject-Model Pairs):** When analyzing all individual data points (8 models × 7 subjects = 56 pairs), the correlation is positive but moderate, suggesting some variance at the specific subject level or potential noise in individual measurements.
  - Pearson r: 0.3833 (p = 0.0035)
  - Spearman $\rho$: 0.2982 (p = 0.0256)
- **Model Mean Performance Correlation:** When analyzing the average performance of each model across all 7 subjects (8 data points), the correlation becomes extremely strong, particularly in terms of rank order. This indicates that while absolute scores differ (YourBench questions are harder), the relative ranking of models is preserved.
  - Pearson r: 0.9646 (p < 0.0001)
  - Spearman $\rho$: 1.0000 (p < 0.0001)

The perfect Spearman correlation for mean model performance strongly supports the validity of YourBench for generating discriminative evaluations that align with established benchmarks in terms of relative model capability assessment.

## F.2 Per-Subject Performance Plots

The following figures visualize the performance comparison for each individual MMLU subject included in the study. Each plot compares the performance of the evaluated LLMs on the original MMLU subset (grey bars) versus the YourBench-generated subset (orange bars). These plots visually complement the aggregated data in Figure 1 and the comprehensive data in Table 2.

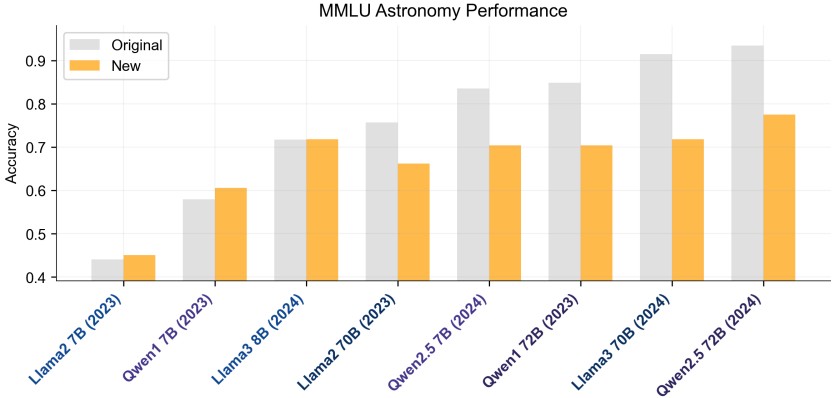

Figure 8: MMLU Replication Performance: Astronomy

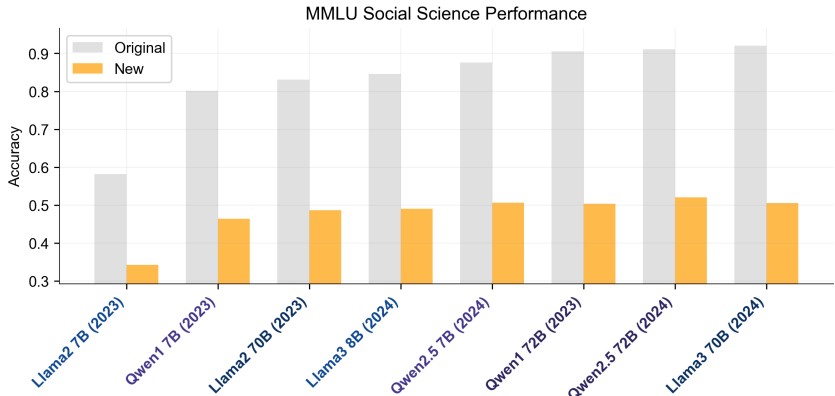

Figure 9: MMLU Replication Performance: Social Science

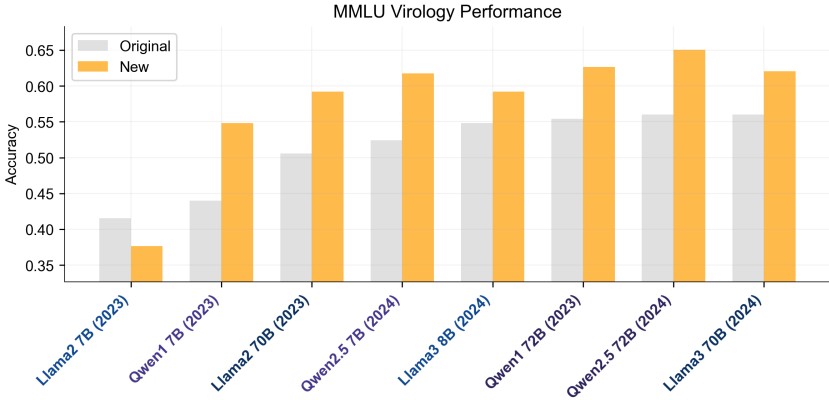

Figure 10: MMLU Replication Performance: Virology

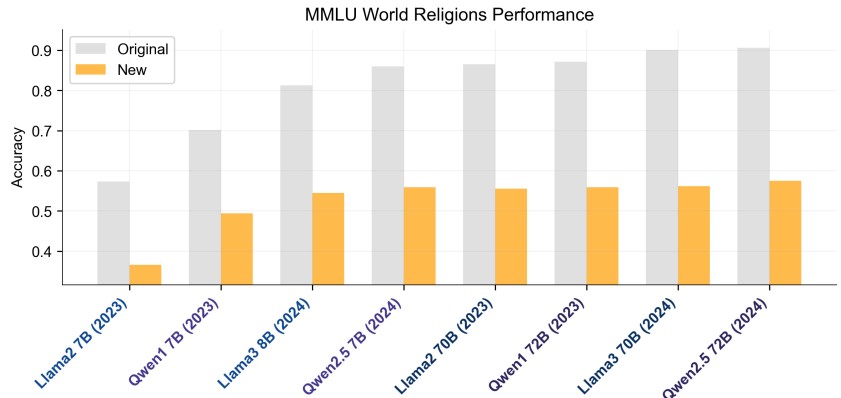

Figure 11: MMLU Replication Performance: World Religions

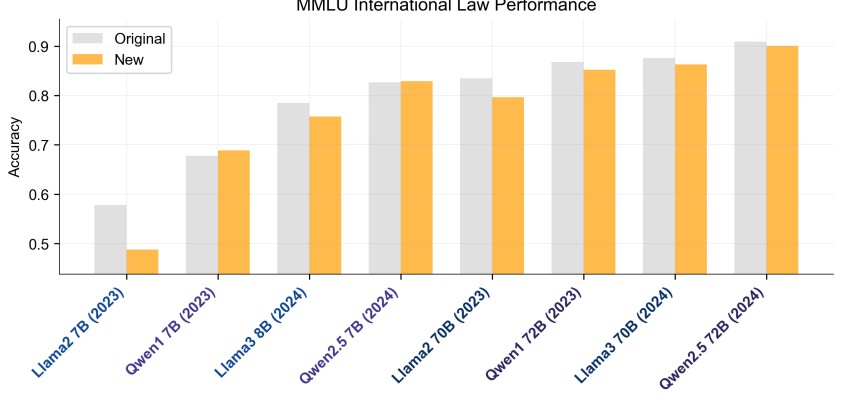

Figure 12: MMLU Replication Performance: International Law

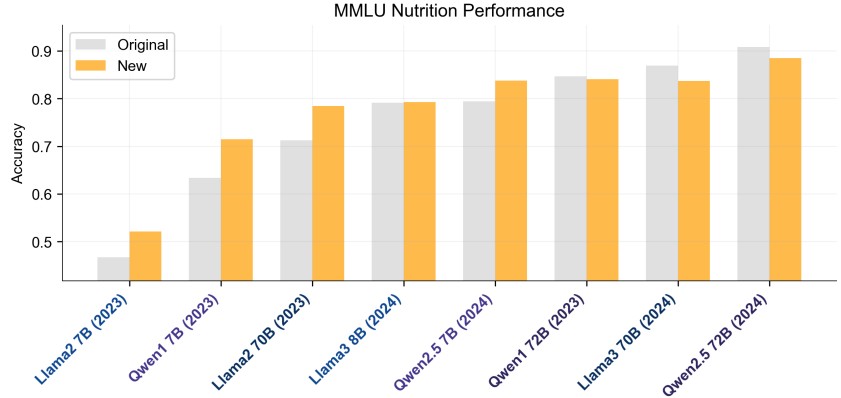

Figure 13: MMLU Replication Performance: Nutrition

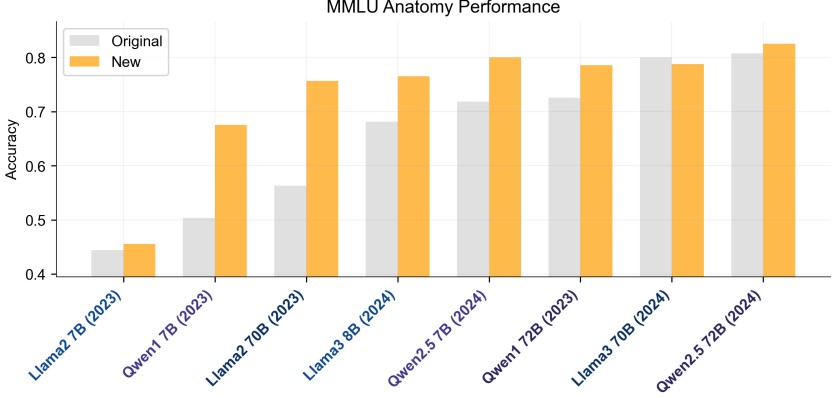

Figure 14: MMLU Replication Performance: Anatomy

### F.3 Comprehensive Performance Table

Table 2 provides the complete numerical results, detailing the accuracy and standard error[5] for each model on both the original ("orig") and YourBench-generated ("new") MMLU subsets across the seven evaluated domains.

Table 2: Comprehensive MMLU Replication Results: Accuracy (Std Err) across Models and Subjects. "New" refers to YourBench-generated benchmarks, "Orig" refers to original MMLU subsets.

| Model | Astronomy | | Social Science | | Virology | | World Religions | |
|---|---|---|---|---|---|---|---|---|
| | New | Orig | New | Orig | New | Orig | New | Orig |
| Qwen1 7B (2023) | 60.56% (5.84%) | 57.89% (4.02%) | 46.37% (1.67%) | 80.10% (2.82%) | 54.82% (1.93%) | 43.98% (3.86%) | 49.43% (1.16%) | 70.18% (3.51%) |
| Qwen2.5 7B (2024) | 70.42% (5.45%) | 83.55% (3.02%) | 50.61% (1.67%) | 87.56% (2.33%) | 61.75% (1.89%) | 52.41% (3.89%) | 55.93% (1.16%) | 85.96% (2.66%) |
| Llama3 8B (2024) | 71.83% (5.38%) | 71.71% (3.67%) | 49.05% (1.67%) | 84.58% (2.55%) | 59.19% (1.91%) | 54.82% (3.87%) | 54.47% (1.16%) | 81.29% (2.99%) |
| Llama2 7B (2023) | 45.07% (5.95%) | 44.08% (4.04%) | 34.19% (1.59%) | 58.21% (3.49%) | 37.65% (1.88%) | 41.57% (3.84%) | 36.60% (1.12%) | 57.31% (3.79%) |
| Llama2 70B (2023) | 66.20% (5.65%) | 75.66% (3.49%) | 48.60% (1.67%) | 83.08% (2.65%) | 59.19% (1.91%) | 50.60% (3.89%) | 55.55% (1.16%) | 86.55% (2.62%) |
| Qwen1 72B (2023) | 70.42% (5.45%) | 84.87% (2.92%) | 50.39% (1.67%) | 90.55% (2.07%) | 62.65% (1.88%) | 55.42% (3.87%) | 55.87% (1.16%) | 87.13% (2.57%) |
| Qwen2.5 72B (2024) | 77.46% (4.99%) | 93.42% (2.02%) | 52.07% (1.67%) | 91.04% (2.02%) | 65.06% (1.85%) | 56.02% (3.86%) | 57.55% (1.15%) | 90.64% (2.23%) |
| Llama3 70B (2024) | 71.83% (5.38%) | 91.45% (2.28%) | 50.50% (1.67%) | 92.04% (1.91%) | 62.05% (1.88%) | 56.02% (3.86%) | 56.15% (1.15%) | 90.06% (2.29%) |

| Model | International Law | | Nutrition | | Anatomy | | Average | |
|---|---|---|---|---|---|---|---|---|
| | New | Orig | New | Orig | New | Orig | New Avg | Orig Avg |
| Qwen1 7B (2023) | 68.87% (1.70%) | 67.77% (4.27%) | 71.45% (1.54%) | 63.40% (2.76%) | 67.57% (2.14%) | 50.37% (4.32%) | 59.87% | 64.80% |
| Qwen2.5 7B (2024) | 82.88% (1.38%) | 82.64% (3.46%) | 83.80% (1.26%) | 79.41% (2.32%) | 80.04% (1.82%) | 71.85% (3.89%) | 70.78% | 78.84% |
| Llama3 8B (2024) | 75.74% (1.57%) | 78.51% (3.75%) | 79.25% (1.39%) | 79.08% (2.33%) | 76.51% (1.94%) | 68.15% (4.02%) | 67.99% | 73.45% |
| Llama2 7B (2023) | 48.79% (1.84%) | 57.85% (4.51%) | 52.10% (1.71%) | 46.73% (2.86%) | 45.53% (2.27%) | 44.44% (4.29%) | 41.41% | 50.03% |
| Llama2 70B (2023) | 79.65% (1.48%) | 83.47% (3.39%) | 78.44% (1.40%) | 71.24% (2.59%) | 75.68% (1.96%) | 56.30% (4.28%) | 67.61% | 72.81% |
| Qwen1 72B (2023) | 85.18% (1.31%) | 86.78% (3.09%) | 84.03% (1.25%) | 84.64% (2.06%) | 78.59% (1.87%) | 72.59% (3.85%) | 69.89% | 79.84% |
| Qwen2.5 72B (2024) | 90.03% (1.10%) | 90.91% (2.62%) | 88.46% (1.09%) | 90.85% (1.65%) | 82.54% (1.73%) | 80.74% (3.41%) | 73.31% | 84.89% |
| Llama3 70B (2024) | 86.25% (1.26%) | 87.60% (3.01%) | 83.68% (1.26%) | 86.93% (1.93%) | 78.79% (1.87%) | 80.00% (3.46%) | 70.61% | 82.01% |

# G   Detailed Related Work and Literature Review

This appendix provides a comprehensive discussion of the related work surveyed in Section 4, detailing the challenges in large language model (LLM) evaluation and prior approaches that motivate the development of YourBench. As models have grown in size and sophistication, traditional evaluation approaches have struggled to keep pace. We survey four key directions in LLM benchmarking—(1) the challenges of static, human-curated benchmarks, (2) synthetic and dynamic benchmark generation, (3) temporal validity concerns, and (4) domain-specific evaluations—and highlight how **YourBench** addresses the major open problems that emerge in each.

## G.1   Limitations of Static Benchmarks

Historically, static benchmarks such as MNIST (Deng, 2012), GLUE (Wang et al., 2019), and SQuAD (Rajpurkar et al., 2016) have been central to measuring progress in machine learning. Although these datasets propelled rapid innovation, modern LLMs can quickly saturate their performance ceilings, sometimes surpassing human-level scores within mere months (Ruder, 2023; Wei, 2023). This *benchmark saturation* hampers their long-term utility in discriminating genuinely more capable models. For instance, models that reached near-perfect scores on GLUE soon forced the community to adopt other, more challenging tasks (Wei, 2023).

An additional concern is *benchmark contamination*, where test data is inadvertently included in a model's training corpus. Because large-scale pretraining involves ingesting vast amounts of web content, popular benchmarks are often seen—or memorized—by the model (Kiela et al., 2021; Ruder, 2023; Zhang et al., 2024). Empirical analyses show that certain LLMs can repeat verbatim segments from question banks such as GSM8K (Cobbe et al., 2021) or MATH (Hendrycks et al., 2021b) when tested in a zero-shot setting (Wei, 2023), artificially inflating performance. Holding out an unseen test set is one partial solution, but as time passes and these datasets spread online, the likelihood of contamination grows (Gupta et al., 2024). Consequently, reliance on a single, static, and publicly available

---

[5]Standard error was derived directly from the accuracy mean, following the methodology in (Fourrier et al., 2023).

benchmark may induce narrow optimization rather than robust generalization (Hendrycks et al., 2021a).

## G.2 Toward Dynamic and Synthetic Evaluation

Faced with saturation and contamination, researchers have pursued *dynamic* and *synthetic* benchmark generation. Kiela et al. (2021) introduced Dynabench to update evaluation sets interactively, challenging models with adversarially crafted queries. This iterative approach demonstrated that once a model adapts to a static test, new data can still reveal surprising failures. However, such human-in-the-loop curation remains expensive and slow to scale.

A more automated strategy is to use LLMs themselves for benchmark synthesis. Several techniques involve prompting a strong generator model to create new questions or tasks, sometimes based on existing ones (*benchmark rewriting*) (Wei, 2023; Krishna et al., 2024). Methods like Auto-Dataset (Ruder, 2023) or ITD (Wei, 2023) rephrase, expand, or mutate original items while controlling for difficulty, ensuring the new tasks remain answerable. Others adopt *multi-agent* pipelines, in which distinct LLMs generate candidate questions and validate them, filtering out ambiguous or erroneous samples (Zhou et al., 2025). Further exploring the role of LLMs in the evaluation pipeline, early work by Shashidhar et al. (2023) utilized LLMs as judges to assess model outputs, correcting for positional bias inherent in such automated evaluations. Despite promising progress, fully synthetic benchmarks introduce new challenges, including the risk of hallucinated or trivial questions. Quality control and verification remain active research topics, especially when the aim is to test advanced reasoning or domain-specific knowledge.

## G.3 Temporal Validity and Knowledge Evolution

Another major challenge is *temporal validity*, reflecting the fact that knowledge and world events change continuously. Many popular benchmarks capture only static snapshots, making them less relevant when facts become outdated (Zhu et al., 2023; Deng et al., 2024). LLM performance thus appears high on older queries but may degrade sharply on newly introduced or time-sensitive questions (Zhu et al., 2023). Holding out a private test set of recent data can help, but frequent refreshes are necessary to track a model's ability to integrate new information (Ruder, 2023; Zhang et al., 2024).

Several works illustrate the severity of the problem. Zhu et al. (2023) generated *post-training* news-based questions to measure whether an LLM truly updates its internal knowledge representation. They found LLMs frequently defaulted to outdated responses, highlighting a gap between real-time information usage and parametric memory. Similarly, Deng et al. (2024) created an evolving dataset of newly coined terminology, demonstrating 20%+ accuracy drops for concepts introduced long after a model's pretraining cutoff. These findings underscore the necessity for *continually updated* benchmarks that can test a model's recency-awareness and its ability to override memorized facts.

## G.4 Domain-Specific Evaluation

Moving from general-purpose benchmarks to specialized ones is increasingly essential, especially in high-stakes fields like medicine, law, and finance (Hung et al., 2023a). Benchmarks such as USMLE-based medical QA (Nori et al., 2023), or specialized legal datasets like Case-HOLD and LegalBench (Holzenkamp et al., 2023), have revealed critical blind spots in LLM reasoning (Hung et al., 2023b). For instance, LLMs might achieve near-human scores on open-domain quizzes yet commit severe factual errors or hallucinations in domain-specific contexts (Gupta et al., 2024).

Building domain-specific benchmarks demands costly expert annotations and must reflect the latest regulations, guidelines, or terminology. In medicine, for example, clinical protocols can change frequently, making a static test rapidly obsolete. Researchers have thus proposed *rolling* domain benchmarks—continuously collected or synthesized data for niche areas such as real-time medical literature or changing legal precedents (Zhang et al., 2024). So far, these dynamic domain evaluations remain nascent: they are typically narrow, small in

size, and do not integrate robust automated generation pipelines or multi-modal content ingestion.

Synthesizing these research themes reveals persistent open problems in LLM benchmarking. **First**, existing static benchmarks are prone to contamination and rapid saturation. **Second**, purely human-driven dynamic approaches cannot scale indefinitely. **Third**, synthetic generation requires careful quality control and can still produce stale or trivial tasks if not refreshed in tandem with new knowledge sources. **Fourth**, few existing solutions integrate domain expertise in a flexible manner or support continuous updates for specialized fields. **Finally**, temporal drift in factual knowledge remains inadequately addressed, as most benchmarks do not systematically ensure that test data are *entirely* post-training or reflective of newly emerging concepts.

# H   Prompts

## H.1   Document Summarization Prompt

The following prompt is first provided into the language model. Once the model provides a response answer, we extract the content that is contained within the final_summary XML tags to function as our document summary.

---

```
You are an AI assistant tasked with analyzing and summarizing documents from various
↪   domains. Your goal is to generate a concise yet comprehensive summary of the given
↪   document. Follow these steps carefully:

1. You will be provided with a document extracted from a website. This document may
↪   contain unnecessary artifacts such as links, HTML tags, or other web-related
↪   elements.

2. Here is the document to be summarized:
<document>
{document}
</document>

3. Before generating the summary, use a mental scratchpad to take notes as you read
↪   through the document. Enclose your notes within <scratchpad> tags. For example:

<scratchpad>
- Main topic: [Note the main subject of the document]
- Key points: [List important information]
- Structure: [Note how the document is organized]
- Potential artifacts to ignore: [List any web-related elements that should be
↪   disregarded]
</scratchpad>

4. As you analyze the document:
    - Focus solely on the content, ignoring any unnecessary web-related elements.
    - Identify the main topic and key points.
    - Note any important details, facts, or arguments presented.
    - Pay attention to the overall structure and flow of the document.

5. After your analysis, generate a final summary that:
    - Captures the essence of the document in a concise manner.
    - Includes the main topic and key points.
    - Presents information in a logical and coherent order.
```

- Is comprehensive yet concise, typically ranging from 3-5 sentences (unless the
↪  document is particularly long or complex).

**6.** Enclose your final summary within <final_summary> tags. For example:

<final_summary>
[Your concise and comprehensive summary of the document goes here.]
</final_summary>

Remember, your task is to provide a clear, accurate, and concise summary of the
↪  document's content, disregarding any web-related artifacts or unnecessary elements.

## H.2   Single Shot Question Generation Prompt

**## Your Role**

You are an expert educational content creator specializing in crafting thoughtful, rich,
↪  and engaging questions based on provided textual information. Your goal is to produce
↪  meaningful, moderately challenging question-answer pairs that encourage reflection,
↪  insight, and nuanced understanding, tailored specifically according to provided
↪  instructions.

**## Input Structure**

Your input consists of:

<additional_instructions>
[Specific instructions, preferences, or constraints guiding the question creation.]
</additional_instructions>

<title>
[Document title]
</title>

<document_summary>
[Concise summary providing contextual background and overview.]
</document_summary>

<text_chunk>
[The single text segment to analyze.]
</text_chunk>

**## Primary Objective**

Your goal is to generate a thoughtful set of question-answer pairs from a single provided
↪  `<text_chunk>`. Aim for moderate complexity that encourages learners to deeply
↪  engage with the content, critically reflect on implications, and clearly demonstrate
↪  their understanding.

**### Context Fields:**

- `<title>`: Contextualizes the content.

- `<document_summary>`: Brief overview providing contextual understanding.
- `<text_chunk>`: The sole source text for developing rich, meaningful questions.
- `<additional_instructions>`: Instructions that influence question style, content, and
  ↪  complexity.

## Analysis Phase

Conduct careful analysis within `<document_analysis>` XML tags, following these steps:

1. **Thoughtful Content Examination**
   - Carefully analyze the given text_chunk, identifying central ideas, nuanced themes,
     ↪  and significant relationships within it.

2. **Concept Exploration**
   - Consider implicit assumptions, subtle details, underlying theories, and potential
     ↪  applications of the provided information.

3. **Strategic Complexity Calibration**
   - Thoughtfully rate difficulty (1-10), ensuring moderate complexity aligned with the
     ↪  additional instructions provided.

4. **Intentional Question Planning**
   - Plan how questions can invite deeper understanding, meaningful reflection, or
     ↪  critical engagement, ensuring each question is purposeful.

## Additional Instructions for Handling Irrelevant or Bogus Information

### Identification and Ignoring of Irrelevant Information:

- **Irrelevant Elements:** Explicitly disregard hyperlinks, advertisements, headers,
  ↪  footers, navigation menus, disclaimers, social media buttons, or any content clearly
  ↪  irrelevant or external to the core information of the text chunk.
- **Bogus Information:** Detect and exclude any information that appears nonsensical or
  ↪  disconnected from the primary subject matter.

### Decision Criteria for Question Generation:

- **Meaningful Content Requirement:** Only generate questions if the provided
  ↪  `<text_chunk>` contains meaningful, coherent, and educationally valuable content.
- **Complete Irrelevance:** If the entire `<text_chunk>` consists exclusively of
  ↪  irrelevant, promotional, web navigation, footer, header, or non-informational text,
  ↪  explicitly state this in your analysis and do NOT produce any question-answer pairs.

### Documentation in Analysis:

- Clearly document the rationale in the `<document_analysis>` tags when identifying
  ↪  irrelevant or bogus content, explaining your reasons for exclusion or inclusion
  ↪  decisions.
- Briefly justify any decision NOT to generate questions due to irrelevance or poor
  ↪  quality content.

## Question Generation Guidelines

### Encouraged Question Characteristics:

- **Thoughtful Engagement**: Prioritize creating questions that inspire deeper thought
  ↪ and nuanced consideration.
- **Moderate Complexity**: Develop questions that challenge learners appropriately
  ↪ without overwhelming them, following the provided additional instructions.
- **Self-contained Clarity**: Questions and answers should contain sufficient context,
  ↪ clearly understandable independently of external references.
- **Educational Impact**: Ensure clear pedagogical value, reflecting meaningful
  ↪ objectives and genuine content comprehension.
- **Conversational Tone**: Formulate engaging, natural, and realistic questions
  ↪ appropriate to the instructional guidelines.

### Permitted Question Types:

- Analytical
- Application-based
- Clarification
- Counterfactual
- Conceptual
- True-False
- Factual
- Open-ended
- False-premise
- Edge-case

(You do not need to use every question type, only those naturally fitting the content and
  ↪ instructions.)

## Output Structure

Present your final output as JSON objects strictly adhering to this Pydantic model within
  ↪ `<output_json>` XML tags:

```python
class QuestionAnswerPair(BaseModel):
    thought_process: str # Clear, detailed rationale for selecting question and analysis
        ↪ approach
    question_type: Literal["analytical", "application-based", "clarification",
                           "counterfactual", "conceptual", "true-false",
                           "factual", "open-ended", "false-premise", "edge-case"]
    question: str
    answer: str
    estimated_difficulty: int  # 1-10, calibrated according to additional instructions
    citations: List[str]  # Direct quotes from the text_chunk supporting the answer
```

## Output Format

Begin by thoughtfully analyzing the provided text_chunk within `<document_analysis>` XML
  ↪ tags. Then present the resulting JSON-formatted QuestionAnswerPairs clearly within
  ↪ `<output_json>` XML tags.

## Important Notes

- Strive to generate questions that inspire genuine curiosity, reflection, and
  ↪  thoughtful engagement.
- Maintain clear, direct, and accurate citations drawn verbatim from the provided
  ↪  text_chunk.
- Ensure complexity and depth reflect thoughtful moderation as guided by the additional
  ↪  instructions.
- Each "thought_process" should reflect careful consideration and reasoning behind your
  ↪  question selection.
- Ensure rigorous adherence to JSON formatting and the provided Pydantic validation
  ↪  model.
- When generating questions, NEVER include phrases like 'as per the text,' 'according to
  ↪  the document,' or any similar explicit references. Questions should inherently
  ↪  integrate content naturally and stand independently without explicit references to
  ↪  the source material

## H.3  Multi Hop Question Generation Prompt

```
## Your Role

You are an expert educational content creator specialized in generating insightful and
  ↪  thoughtfully designed multi-hop questions. Your task is to craft sophisticated,
  ↪  moderately challenging questions that inherently require careful, integrative
  ↪  reasoning over multiple chunks of textual information. Aim to provoke thoughtful
  ↪  reflection, nuanced understanding, and synthesis, particularly when the provided
  ↪  text allows for it.

## Input Structure

Your input will consist of these components:

<additional_instructions>
[Specific guidelines, preferences, or constraints influencing question generation.]
</additional_instructions>

<title>
[Document title]
</title>

<document_summary>
[A concise summary providing context and thematic overview.]
</document_summary>

<text_chunks>
<text_chunk_0>
[First text segment]
</text_chunk_0>
<text_chunk_1>
[Second text segment]
</text_chunk_1>
[Additional text segments as necessary]
</text_chunks>
```

## Primary Objective

Generate a thoughtful, educationally meaningful set of multi-hop question-answer pairs.
↪  Questions should ideally integrate concepts across multiple text chunks, challenging
↪  learners moderately and encouraging critical thinking and deeper understanding.

### Context Fields:
- `<title>`: Document context
- `<document_summary>`: Broad contextual summary for orientation
- `<text_chunks>`: Source material to form integrative multi-hop questions
- `<additional_instructions>`: Specific instructions guiding the complexity and depth of
↪  questions

## Analysis Phase

Perform careful analysis within `<document_analysis>` XML tags:

1. **In-depth Text Analysis**
   - Thoughtfully read each text chunk.
   - Identify key themes, nuanced details, and subtle connections.
   - Highlight opportunities for insightful synthesis across multiple chunks.

2. **Reasoning Path Construction**
   - Construct potential pathways of multi-hop reasoning by connecting ideas, details, or
     ↪  implications found across text chunks.

3. **Complexity Calibration**
   - Rate difficulty thoughtfully on a scale of 1-10, moderately challenging learners
     ↪  according to provided additional instructions.

4. **Strategic Question Selection**
   - Choose questions that naturally emerge from the depth and complexity of the content
     ↪  provided, prioritizing integrative reasoning and genuine curiosity.

## Question Generation Guidelines

### Question Characteristics
- **Multi-Hop Integration**: Questions should naturally require integration across
↪  multiple chunks, demonstrating clear interconnected reasoning.
- **Thoughtfulness & Complexity**: Construct questions that stimulate critical thinking,
↪  reflection, or moderate challenge appropriate to the content.
- **Clarity & Precision**: Ensure each question and answer clearly and concisely
↪  communicates intent without ambiguity.
- **Educational Relevance**: Ensure each question has clear pedagogical purpose,
↪  enhancing understanding or critical reflection.
- **Authentic Language**: Use engaging, conversational language reflecting genuine human
↪  curiosity and inquiry.

### Suggested Question Types
(Use naturally, as fitting to the content complexity)
- Analytical
- Application-based
- Clarification

- Counterfactual
- Conceptual
- True-False
- Factual
- Open-ended
- False-premise
- Edge-case

## **Filtering Irrelevant Content**:
  - **Ignore completely** any irrelevant, redundant, promotional, or unrelated content,
  ↪  including headers, footers, navigation links, promotional materials, ads, or
  ↪  extraneous hyperlinks frequently found in web extracts.
  - **Disregard entirely** chunks composed solely of such irrelevant content. Do **not**
  ↪  generate questions from these chunks.
  - When partially relevant content is mixed with irrelevant material within the same
  ↪  chunk, carefully extract only the meaningful, educationally relevant portions for
  ↪  your integrative analysis.

- **Evaluating Chunk Quality**:
  - If, upon careful analysis, a chunk does not provide sufficient meaningful context or
  ↪  substantial educational relevance, explicitly note this in the
  ↪  `<document_analysis>` section and refrain from generating questions based on it.

- **Prioritizing Quality and Relevance**:
  - Always prioritize the quality, clarity, and educational integrity of generated
  ↪  questions. Do not force questions from unsuitable content.

## Output Structure

Present output as JSON objects conforming strictly to the following Pydantic model within
↪  `<output_json>` XML tags:

```python
class QuestionAnswerPair(BaseModel):
    thought_process: str # Explanation of integrative reasoning and rationale
    question_type: Literal["analytical", "application-based", "clarification",
                           "counterfactual", "conceptual", "true-false",
                           "factual", "open-ended", "false-premise", "edge-case"]
    question: str
    answer: str
    estimated_difficulty: int  # 1-10, moderately challenging as per additional
    ↪  instructions
    citations: List[str]  # Exact supporting quotes from text_chunks
```

## Output Format

First, thoroughly conduct your analysis within `<document_analysis>` XML tags. Then,
↪  provide your synthesized question-answer pairs as valid JSON within `<output_json>`
↪  tags.

## Important Notes

- Prioritize depth and thoughtfulness in your reasoning paths.
- Allow natural complexity to guide question formulation, aiming for moderate challenge.
- Precisely cite verbatim excerpts from text chunks.
- Clearly communicate your thought process for integrative reasoning.
- Adhere strictly to JSON formatting and Pydantic validation requirements.
- Generate questions that genuinely inspire deeper reflection or meaningful exploration
↪  of the provided content.
- When generating questions, NEVER include phrases like 'as per the text,' 'according to
↪  the document,' or any similar explicit references. Questions should inherently
↪  integrate content naturally and stand independently without explicit references to
↪  the source material

## H.4   Judge System Prompt

You will be provided with the summary of a document, a piece of text, a question
↪  generated from that text, and the correct or "gold" answer to the question.
↪  Additionally, you will receive two answers: Answer A and Answer B. Your task is to
↪  determine which of these answers is closer to the gold answer by assessing the
↪  overlap of key points between the ground truth and the two given answers.

# Steps

1. **Document Understanding**:
   - Analyze the provided document summary to grasp the context and main themes.

2. **Chunk Understanding**:
   - Examine the provided text (chunk) to understand its content.

3. **Question Understanding**:
   - Interpret the given question to fully comprehend what is being asked.

4. **Ground Truth Answer Understanding**:
   - Understand the provided ground truth answer, identifying its key points.

5. **Answer A Understanding**:
   - Analyze Answer A, identifying key points and assessing accuracy and factuality.

6. **Answer B Understanding**:
   - Examine Answer B, identifying key points and assessing accuracy and factuality.

7. **Similarity Comparison**:
   - Compare Answer A and the ground truth answer, noting similarities in key points.
   - Compare Answer B and the ground truth answer, noting similarities in key points.

8. **Final Similarity Analysis**:
   - Evaluate both answers based on the similarities identified and determine which is
     ↪  closer to the ground truth in terms of key points and factuality.

# Output Format

- Provide your final evaluation of which answer is closer to the ground truth within
↪  `<final_answer>` XML tags.

**-** Include a detailed analysis for each part within the designated XML tags:
↪   `<document_understanding>`, `<chunk_understanding>`, `<question_understanding>`,
↪   `<ground_truth_answer_understanding>`, `<answer_a_understanding>`,
↪   `<answer_b_understanding>`, `<similarity_comparison_answer_a>`,
↪   `<similarity_comparison_answer_b>`, and `<final_similarity_analysis>`.

# Examples

**Input**:
```xml
<document_summary>
[Summary]
</document_summary>

<piece_of_text>
[Text]
</piece_of_text>

<question>
[Question]
</question>

<gold_answer>
[Gold Answer]
</gold_answer>

<answer_a>
[Answer A]
</answer_a>

<answer_b>
[Answer B]
</answer_b>
```
**Output**:
```xml

<document_understanding>
Understanding of the summary including key themes
</document_understanding>

<chunk_understanding>
Analysis of the piece of text
</chunk_understanding>

<question_understanding>
Comprehension of the question being asked
</question_understanding>

<ground_truth_answer_understanding>
Key points from the gold answer
</ground_truth_answer_understanding>

<answer_a_understanding>
```

```
Key points and accuracy of Answer A
</answer_a_understanding>

<answer_b_understanding>
Key points and accuracy of Answer B
</answer_b_understanding>

<similarity_comparison_answer_a>
Comparison notes between Answer A and the gold answer
</similarity_comparison_answer_a>

<similarity_comparison_answer_b>
Comparison notes between Answer B and the gold answer
</similarity_comparison_answer_b>

<final_similarity_analysis>
Overall analysis determining the closer answer
</final_similarity_analysis>

<final_answer>
Answer X (where X is the option you pick)
</final_answer>
```

# Notes

- Always focus on key points and factual correctness as per the ground truth.
- Avoid any biases and rely solely on the evidence presented.
- Enclose all evaluations and analyses in the specified XML tags for clarity and
  ↪  structure.

# I    Question Validity

## I.1    Valid Question Examples

### I.1.1    Example 1

```
# Question Details
## Source Information

iraqi immigrant hailed as hero for preventing armed robbery at ypsilanti juice shop
↪    ypsilanti, mich. (wxyz) — vara juice in ypsilanti nearly became the victim of an
↪    armed robbery this past friday. caught on camera, the suspect had no clue that his
↪    attempt to make quick cash would come to a hard stop, all thanks to a hero who was
↪    next door. thirty-five-year-old ali hadma owns a hookah place called cups on a
↪    mission, located next to vara juice on washtenaw ave. **"3 years,"** said ali when
↪    asked how long he's owned the shop. ali pins the suspect against the counter. a
↪    struggle to control the firearm begins. ali disarms the suspect. and eventually takes
↪    him down. "have you got any tactical or self-defense training? " i asked. "no. i just
↪    go to the gym 6 days a week," said ali. once ali got the cash back, he let go of the
↪    suspect, who can be seen walking away in the security footage. all the girls he
↪    treats like his sisters,"** said sadam badani, the owner of the vara juice location.
↪    badani tells me mariam is doing okay, but her parents will only allow mariam to
↪    resume work if her hero, ali, is around. "i don't care about the money, about
↪    anything else. as long as nobody got hurt," said sadam. "whenever ali need me, i'll
↪    be there," said sadam.

## Question

In what ways have Ali's actions during the robbery influenced the community's perception
↪    of him and their sense of security?

## Answer

Ali's actions during the robbery have made him a local hero and gained him widespread
↪    appreciation. The community, including the juice shop owner and employees, deeply
↪    appreciates his bravery and quick thinking. This has led to a stronger sense of
↪    security, with the juice shop owner stating that Mariam can only resume work if Ali
↪    is around.

## Citations

[All the girls he treats like his sisters," said Sadam Badani, the owner of the Vara
↪    Juice location.,"Whenever Ali need me, I'll be there," said Sadam.]

# Human Evaluation

## Determination

valid

## Reasoning

-
```

# Generation Details

## Model

mistralai/Mistral-Large-Instruct-2411

## Question Category

open-ended

## Kind

multi_hop

## Estimated Difficulty

6/10

---

*I.1.2   Example 2*

---

# Question Details
## Source Information

(truncated)...   (pn12-36) christopher landau (cal. no. 41) (pn12-25) ordered, that
↪   following the conclusion of morning business on monday, march 24, 2025, the senate
↪   proceed to executive session and resume consideration of the nomination of john
↪   phelan, of florida, to be secretary of the navy. (mar. 14, 2025. ) michael kratsios
↪   (cal. no. 38) (pn13-8) jayanta bhattacharya (cal. no. 44) (pn12-2) martin makary
↪   (cal. no. 45) (pn12-28) james bishop (cal. no. 39) (pn12-3) aaron reitz (cal. no. 48)
↪   (pn12-37) ordered, that on tuesday, march 25, 2025, the cloture motions on the
↪   following nominations ripen: michael kratsios, of south carolina, to be director of
↪   the office of science and technology policy; jayanta bhattacharya, of california, to
↪   be director of the national institutes of health; martin makary, of virginia, to be
↪   commissioner of food and drugs, department of health and human services; james
↪   bishop, of north carolina, to be deputy director of the office of management and
↪   budget; and aaron reitz, of texas, to be an assistant attorney general. * 33 25-32
↪   jonathan mckernan, of tennessee, to be mar 06, 2025 reported by mr. director, bureau
↪   of consumer financial protection for a term of five years, vice rohit chopra. scott
↪   sc, committee on banking, housing, and urban affairs, without printed report.
↪   department of defense * 36 12-36 john phelan, of florida, to be secretary of the mar
↪   11, 2025 reported by mr. navy, vice carlos del toro, resigned. wicker, committee on
↪   armed services, without printed report. mar 12, 2025 reported by mr. risch, committee
↪   on foreign relations, without printed report. department of veterans affairs * 43
↪   13-9 paul lawrence, of virginia, to be deputy mar 12, 2025 reported by mr. secretary
↪   of veterans affairs, vice tanya j. bradsher, resigned. moran, committee on veterans'
↪   affairs, without printed report. * signifies nominee's commitment to respond to
↪   requests to appear and testify before any duly constituted committee of the senate
    5 nominations calendar no. mar 13, 2025 reported by mr. grassley, committee on the
↪   judiciary, without printed report. mar 13, 2025 reported by mr. grassley, committee
↪   on the judiciary, without printed report. mar 13, 2025 reported by mr. grassley,
↪   committee on the judiciary, without printed report. mar 13, 2025 reported by mrs.
↪   capito, committee on environment and public works, without printed report. * 50 25-53
↪   aaron szabo, of virginia, to be an assistant mar 13, 2025 reported by mrs

## Question

On what date are cloture motions for the nominations of Michael Kratsios, Jayanta
↪  Bhattacharya, Martin Makary, James Bishop, and Aaron Reitz set to ripen, and what are
↪  their respective positions?

## Answer

The cloture motions for Michael Kratsios (Director of the Office of Science and
↪  Technology Policy), Jayanta Bhattacharya (Director of the National Institutes of
↪  Health), Martin Makary (Commissioner of Food and Drugs, Department of Health and
↪  Human Services), James Bishop (Deputy Director of the Office of Management and
↪  Budget), and Aaron Reitz (Assistant Attorney General) are set to ripen on Tuesday,
↪  March 25, 2025.

## Citations

['Mar. 14, 2025. Ordered, That on Tuesday, March 25, 2025, the cloture motions on the
↪  following nominations ripen: Michael Kratsios, of South Carolina, to be Director of
↪  the Office of Science and Technology Policy; Jayanta Bhattacharya, of California, to
↪  be Director of the National Institutes of Health; Martin Makary, of Virginia, to be
↪  Commissioner of Food and Drugs, Department of Health and Human Services; James
↪  Bishop, of North Carolina, to be Deputy Director of the Office of Management and
↪  Budget; and Aaron Reitz, of Texas, to be an Assistant Attorney General.']

# Human Evaluation

## Determination

Valid

## Reasoning

question, answer and citations are correct

# Generation Details

## Model

Qwen/Qwen2.5-14B-Instruct

## Question Category

factual

## Kind

multi-hop

## Estimated Difficulty

*I.1.3  Example 3*

# Question Details
## Source Information

org. following the selection process, all applications will be destroyed. questions?
↪  please send an email to: scholarships@agbell. org response time may be up to three
↪  business days, so please plan accordingly when submitting your questions. george h.
↪  nofer scholarship for law 2025 please type or print clearly and review for accuracy;
↪  illegible or incorrect information will delay review and could disqualify your
↪  application. identifying information name (first, mi, last):
↪  ____________________________________________________________ date of birth
↪  (mm/dd/yyyy) __________ gender: male female complete mailing address:
↪  _________________________________________________________ email address:
↪  ______________________________________________________________
↪  communication throughout the process will be via email. if you do not provide an
↪  email address, if it is written incorrectly, or if we are not able to read it, we
↪  will not be able to communicate with you. telephone number: ______________________
↪  hearing health history age when hearing loss was diagnosed: __________ *if you do not
↪  have a cochlear implant and your pta is below 60db in your better-hearing ear, you do
↪  not qualify.
## Question

How will applicants be contacted regarding updates or decisions about their scholarship
↪  application?

## Answer

Communication throughout the process will be via email.

## Citations

['Communication throughout the process will be via email.']

# Human Evaluation

## Determination

valid

## Reasoning

-

# Generation Details

## Model

google/gemini-2.0-flash-001

## Question Category

factual

## Kind

single shot

## Estimated Difficulty

6/10

---

## I.2 Invalid Question Examples

### I.2.1 Example 1

---

# Question Details
## Source Information

according to the committee, out of the 40 who signed up to deliver testimony, 38 were
↪   opposed to the bill. one of the biggest points of concern was in relation to the
↪   staff-to-child ratio being lowered. as the bill is currently written, a single person
↪   would be allowed to run a large daycare facility overseeing many children. those in
↪   support of the bill believe that won't be a problem and instead, will open up more
↪   opportunities for idahoans to start daycare businesses of their own. chris cargill
↪   with mountain states policy center explained, "we believe that if the legislation is
↪   passed, we will see an increase in new and quality childcare providers in idaho. "
↪   mark kirby of soda springs, idaho, told the tragic story of how his 11-week-old
↪   nephew lost his life after being dropped off at a daycare. "later that afternoon, she
↪   got a call stating that something was very wrong. upon arriving there, she was
↪   escorted in and learned that her son had passed away from asphyxiation," said a
↪   teary-eyed kirby. kirby explained that an investigation found the death was due to
↪   overwhelming child-to-staff ratios and negligence of supervision while the baby was
↪   napping. the committee passed the bill five to four. "we are clearly disappointed
↪   with the outcome," said kirby. "i feel more disappointed about losing this bill than
↪   any other in the session. we have worked on this issue for decades to ensure that we
↪   are promoting business practices and children's safety," added senator melissa
↪   wintrow. the bill will now head to the senate floor. if a majority of state senators
↪   vote to pass the legislation, it will head to the governor's desk, where it will
↪   either be vetoed or signed into law. this is a developing story, and we will continue
↪   to update it as more information becomes available. ## more news in downtown boise we
↪   cover stories making an impact in downtown boise. this is your home to stay on top of
↪   what is changing in downtown boise and why it matters to you and your family. we want
↪   to hear from you! and tell us what we should be covering in your neighborhood.

## Question

How did the death of a child at a daycare facility through asphyxiation influence the
↪   testimony given during the committee hearing on House Bill 243?

## Answer

The tragic death caused by asphyxiation due to high child-to-staff ratios was a pivotal
↪   moment that resonated emotionally with the committee members and played a
↪   significant role in bolstering the opposition to the bill, underscoring safety
↪   concerns related to its proposed deregulation.

## Citations

[ "'Later that afternoon, she got a call stating that something was very wrong. Upon
↪   arriving there, she was escorted in and learned that her son had passed away from
↪   asphyxiation,' said a teary-eyed Kirby." ]

# Human Evaluation

## Determination

Invalid

## Reasoning

the citations don't support the answer. it is also factually inaccurate according to the
↪   text

# Generation Details

## Model

microsoft/Phi-4-mini-instruct

## Question Category

Factual

## Kind

multi_hop

## Estimated Difficulty

6

*I.2.2   Example 2*

# Question Details
## Source Information

(truncated)... and trustworthiness. to prepare, many are increasing their cyber budgets
↪  with a particular focus on data protection and trust. by strategically investing in
↪  these areas, companies are not only building resilience but positioning themselves
↪  positively to their customers. ### investing in what matters most: cloud and data
↪  trust go hand-in-hand over the next 12 months, organisations are prioritising data
↪  protection/trust and cloud security above other cyber investments. they understand
↪  that securing sensitive information is vital to maintaining stakeholder trust and
↪  brand integrity. g. , reducing the time to recover mission-critical data or patching
↪  a system). - - determine the business value of data protection and cloud security to
↪  gain stakeholder trust and make more informed cybersecurity investment decisions. - -
↪  collaborate with tech, security and finance executives to pinpoint the most
↪  essential data security and integrity priorities to guide the information and cloud
↪  security investment strategy. confirming data quality and readiness is necessary to
↪  increase security investments. ## is your cyber strategy and leadership driving real
↪  resilience? from lagging resilience efforts to gaps in ciso involvement in strategic
↪  decisions, there are clear areas where strategic alignment is needed. to get there,
↪  organisations should emulate the leading cybersecurity practices of their top
↪  performing peers. they should also move beyond addressing known threats and implement
↪  an agile, secure-by-design approach to business, one that strives to build trust and
↪  lasting resilience. ### partial implementation isn't enough despite mounting
↪  concerns about cyber risk, most businesses are struggling to fully implement cyber
↪  resilience across core practices. a review of 12 resilience actions across people,
↪  processes and technology indicates that 42% or fewer of executives believe their
↪  organisations have fully implemented any one of those actions. more concerning, only
↪  2% say all 12 resilience actions have been implemented across their organisation.
↪  this leaves a glaring vulnerability — without enterprise-wide resilience, companies
↪  remain dangerously exposed to the increasing threats that could compromise the
↪  entire operation. ### elevating the ciso: aligning strategy with security many
↪  organisations miss critical opportunities by not fully involving their cisos in key
↪  initiatives. fewer than half of executives tell us that their cisos are largely
↪  involved in strategic planning for cyber investments, board reporting and overseeing
↪  tech deployments. this gap leaves organisations vulnerable to misaligned strategies
↪  and weaker security postures. ## bridging the gaps to cyber resilience: the c-suite
↪  playbook ### sign up to get the full playbook and access more of the latest findings
↪  for 2025. ### about the survey the 2025 global digital trust insights is a survey of
↪  4,042 business and technology leaders conducted in the may through july 2024 period.
↪  - a quarter of leaders are from large companies with $5 billion or more in revenues.
↪  respondents operate in a range of industries, including industrials and services
↪  (21%), tech, media, telecom (20%), financial services (19%), retail and consumer
↪  markets (17%), energy, utilities, and resources (11%), health (7%) and government
↪  and public services (4%). - respondents are based in 77 countries. the regional
↪  breakdown is western europe (30%), north america (25%), asia pacific (18%), latin
↪  america (12%), central and eastern europe (6%), africa (5%) and middle east (3%).

## Question

What are the critical factors that organizations should prioritize to achieve true cyber
↪  resilience, and how do they interconnect?

## Answer

Critical factors include CISO involvement in strategic planning, adopting a
↪   secure-by-design approach, and investing in data protection and cloud security. These
↪   elements interconnect by ensuring strategic alignment, proactive security measures,
↪   and building stakeholder trust, which collectively enhance organizational resilience.

## Citations

['While the rapid advancement of generative AI (GenAI) is ushering in new opportunities
↪   across industries, it also presents cybersecurity risks.',
'To prepare, many are increasing their cyber budgets with a particular focus on data
↪   protection and trust.',
'They should also move beyond addressing known threats and implement an agile,
↪   secure-by-design approach to business, one that strives to build trust and lasting
↪   resilience.']

# Human Evaluation

## Determination

Invalid

## Reasoning

answer looks to be correct, but the citations are wrong

# Generation Details

## Model

DeepSeek-R1-Distill-Qwen-14B

## Question Category

analytical

## Kind

multi-hop

## Estimated Difficulty

8/10

I.2.3   *Example 3*

# Question Details
## Source Information

"it's nice to have the memories, but i wish we could make more," said jesse collins,
↪ alexis' uncle. **investigative findings:** - **david plagmann, 36**: responsible for
↪ alexis' death. fired by the shelby county sheriff's office. - **marianne and jesse
↪ collins**: alexis' aunt and uncle. - **jake collins**: alexis' father. alexis' family
↪ describes her as having a soft and loving heart, always step up to care for others,
↪ including her four children. she was always positive and believed things would
↪ improve, even if it was hard.

## Question

How many children did Alexis Martin-Collins have?

## Answer

Four children

## Citations

[She was always positive and believed things would improve, even if it was hard.]

# Human Evaluation

## Determination

Invalid

## Reasoning

answer is correct and factual, and it makes a valid citation, but the citation points to
↪ wrong part of text

# Generation Details

## Model

claude-3-5-haiku-20241022

## Question Category

factual

## Kind

single shot

## Estimated Difficulty

2/10

