# OpenReview forum: "Yourbench: Dynamic Evaluation Set Generation with LLMs"
_colmweb.org/COLM/2025/Conference — COLM 2025_

### Official Review · Reviewer_SnWz · 2025-04-30

**Rating:** 7
**Confidence:** 4
**Ethics Flag:** 1

**Summary:**

This work devotes to improve LLM evaluation by proposing YourBench. YourBench is an open-sourced framework that generates evaluation samples dynamically and automatically from user-provided documents. Results show that YourBench not only provides more flexibility but also can make trustworthy conclusions cheaper and faster.

**Reasons To Accept:**

1) LLMs are general-purpose models, but users always have their only use cases. It is hard to figure out a very user-specific benchmark for everyone because it is timely and expensive on human annotation. However, this work propose to generate queries from user-provided documents automatically, which can greatly alleviate this problem.

**Reasons To Reject:**

1) I don't understand why this apporach can really solve contamination problem. It is possible that the user-document is contaminated as well if it has been used in the pretraining.
2) I also don't understand why this approach can workaround saturation. Instead, if my understanding is correct, the question difficulty highly depends the on the source document the the capacity of LLM generating evaluation samples. So it is actually easy to saturate in my opinion.

Although YourBench seems have some weakness, I still believe that the flexibility is important for general-purpose evaluation. Preparing documents are always easier than preparing QA pairs for user. So I tend to accept this work as a good research on LLM eval.

---

> ### Author Response · Authors · 2025-06-02
> **Response to Reviewer SnWz**
>
> We thank the reviewer for their constructive feedback and for recognizing YourBench's importance for flexible, general-purpose evaluation. We address the two main concerns below.
>
> ---
>
> ### **Addressing Contamination**
>
> YourBench addresses contamination through its ability to generate benchmarks from previously unseen datasets, either documents collected after model training cutoffs (e.g., Tempora) or entirely internal/private documents. This fundamentally differs from evaluating on published benchmarks that may have inadvertently contaminated training data.While benchmarks generated with YourBench could become contaminated if published without gating, the key advantage is regeneration capability: new benchmarks can be created extremely inexpensively whenever contamination is suspected, unlike traditional human-annotated benchmarks.
>
> ---
>
> ### **Saturation Mitigation**
>
> The reviewer correctly notes that question difficulty depends on source documents and generator capacity. However, YourBench offers several mitigation strategies:
>
> 1. **Document Selection**: Using complex source documents combined with multi-hop question generation produces inherently challenging questions, as yourbench primarily acts as a scaffold
> 2. **Generation-Answer Gap**: Interestingly, models often cannot answer questions they generate without document access.
>
> To illustrate this second point, we conducted an ablation study using Qwen3-30B-A3B to generate questions from FineWeb-EDU (filtered for scores >4). Despite this corpus likely being in the model's pretraining data, Qwen3-30B-A3B achieved only 75.3% accuracy on its own generated multiple-choice questions (77.4% with reasoning). This suggests that while models may possess the information, it is not sufficiently well-embedded in their latent space to ensure reliable retrieval during evaluation. This generation-answer gap provides a natural buffer against premature saturation, even when using familiar source material.
>
> ---
>
> We hope this clarifies how YourBench addresses these important concerns, and we thank the reviewer for their time!

---

> > ### Comment · Reviewer_SnWz · 2025-06-06
> >
> > Thanks for the clarification.
> > i decided to keep my score.

---

### Official Review · Reviewer_pLCc · 2025-05-11

**Rating:** 2
**Confidence:** 4
**Ethics Flag:** 1

**Summary:**

Main idea: Framework (i.e. pipeline) for guiding language models to construct new benchmarks given minimal supervision

Evidence:
- 7 subsets of MMLU have new replicas generated using YourBench
- One (or two, depending on how one counts) new benchmarks are created to reduce contamination and test new knowledge post March 2025

**Reasons To Accept:**

- This is a compelling problem to work on and solve
- The authors' framework and general overall approach seem sensible, albeit limited to question answering benchmarks

**Reasons To Reject:**

I think this paper is conceptually fine, but needs to do a LOT more work to build trust in this framework/pipeline.

## Section 2

- Section 2.2. I feel like a key criterion is missing from D2EG: importance. The three listed criteria (coverage, diversity, answerability) are all well and good, but content is not created equally and evaluations should concentrate on "the material we care about". One could envision asking questions like "What word does the Nth sentence end on?" or "How many words contain the letter E?" or "How many links are contained in the paragraph", which could meet all three criteria, but aren't really the substance we want to evaluate models on. Maybe a different term should be "external validity", by which I mean: do consumers of models' benchmarks scores feel like the scores are capturing the important elements of the task at hand?
- The anonymized URLs at the bottom of page 2 in the PDF both direct to `https://openreview.net/ANONYMIZED%20URL.pdf#[0,{%22name%22:%22Fit%22}]`. I am thus unable to scrutinize the YourBench bench or the TEMPORA-0325 dataset.
- In Appendix D.2, which LLMs were used in the ensemble of judges? I was unable to find this information.
- I'm not sure I see any analyses about the effects of YourBench's hyperparameters: $\theta_{\text{cit}}$ and $\tau_{\text{sim}}$. How robust is the YourBench framework to these hyperparameters?

## MMLU Replication

- I think many more models should be added to Figure 1. This would significantly strengthen the claim that YourBench's MMLU replica "perfectly preserves the relative performance ranking" and "Spearman ρ=1.00". There are many more than 8 models in the open-parameter ecosystem, and 7 subsets of MMLU is not so expensive to evaluate. For suggestions: DeepSeek, Gemma 1 & 2, Phi, OLMo, Mistral, Pythia, Cerebras, LLM360 https://huggingface.co/LLM360, etc.
- I think Figure 1 would be better visualized as a scatterplot with Original scores on one axis and YourBench Synthesized scores on another axis. I say this for two reasons: (1) the specific models here are less important than preserving the ranking, and (2) as I comment below, preserving the ranking of scores is oftentimes insufficient to evaluate models as one oftentimes wants to know the quantitative difference in performance. One way to do this would be to have a scatterplot per subset, arranged in a 2-rows-by-4-columns grid.
- As an extension of the previous point, here, we actually have 7 different YourBench-constructed benchmark subsets. I feel like Figure 1 should show both an aggregate score (as it currently does) but also a per-subset score. The latter feels even more important to me because if an ordering is preserved for each benchmark subset, then the same ordering is likely to hold for the aggregate score, but the opposite is less likely to be true.
- After making the above point, I realized that the authors must surely have done this analysis. I found Appendix F1 and discovered that Spearman drops from 1.00 to 0.2982. This is also mentioned on Page 8. This is a terrible degradation and I am unhappy that this incongruity is not front and center in Figure 1.
- Relatedly, it is remarkable that Pearson and Spearman correlations are high in aggregate but low for each subset. This feels extremely surprising to me.
- It is unclear to me how many times the authors generated their MMLU replication and/or whether it was "tuned" in any way. Ideally the authors have evidence that the dataset was generated once, and that no hill climbing was performed on the dataset generation methodology. Otherwise, the MMLU replication results are not representative of YourBench. Could the authors please provide such evidence?
- Table 1 in Appendix F3 shows standard errors. I prefer confidence intervals, and such uncertainty should be visualized in the results. For instance, here is one way to combine a scatterplot with uncertainty in Python visualizations https://matplotlib.org/3.1.1/gallery/statistics/errorbar_limits.html
- This may be minor, but which model(s) were included in the LLM ensemble (Page 4) used to generate the MMLU Replication Dataset? This seems like a key consideration. I was unable to find this information. I'll call these the dataset-generating LLMs.

## Transcending Dataset-Generating LLMs

- A key question that I don't see addressed is whether YourBench offers a means to evaluate new frontier models. That is, if a benchmark is generated by LLM X, and LLM Y is widely regarded as superior to X, is the benchmark capable of accurately evaluating Y? Or will Y trivially ace the benchmark because anything X can do, Y can do better?

## Appendix F

- Please add the Pearson and Spearman correlations to each figure caption (Figures 9 through 15). As with Figure 1, I would also prefer more models to be added and the data visualized as a scatterplot, but the authors are welcome to push back.

---

> ### Author Response · Authors · 2025-05-29
> **Response to Reviewer pLCc**
>
> We sincerely thank the reviewer for their thoughtful comments. We are encouraged that they find the problem compelling and our approach sensible. Below we address the main questions systematically.
>
> # **Importance Criterion, Question Quality**
>
> The "importance" criterion is indeed a primordial criteria, that we have embedded implicitly in our framework through multiple mechanisms: (1) Our generation prompts explicitly instruct models to focus only "meaningful content" (pp.33,37), (2) System prompts are curated for "educational importance" (Appendix H), and (3) Human evaluation confirmed ~85% validity across 2k+ questions (Appendix E.1), with the frontier models scoring at 96%. We have also performed manual review to ensure that the questions are not trivial or irrelevant. We will therefore add it explicitly to the formalisation as the 4th D2EG criterion and thank the reviewer for their suggestion.
>
> ---
>
> # **Anonymous Links**
>
> The benchmarking code was provided in the supplementary materials. Due to the size of the datasets, they were not attached to the supplements, and we had anonymised the links for the double blind policy. We have made the data available anonymously:
>
> - Tempora: https://huggingface.co/datasets/anonymous-paper-author/tempora
> - Yourbench Traces: https://huggingface.co/datasets/anonymous-paper-author/tempora_yourbench_traces
> - MMLU replication: https://huggingface.co/collections/anonymous-paper-author/mmlu-reproduction-studies-68372c393377d718802415ee
>
> We trust that this will not only address the reviewer's concern of not being able to view the dataset, but also concerns regarding question quality
>
> ---
>
> # **Correlations**
>
> The reviewer correctly identifies an important nuance, which is that while YourBench preserves model rankings on the full dataset (capturing a general hierarchy of model capability), it exhibits natural variance on the individual subject-model pairs. This variance is due to sampling effects, linked to the overall small size of the original subsets: they contain between 100 and 400 samples, and we observe that the larger the original subset (more stable), the stronger the correlation with our (bigger) reproductions.
>
> ---
>
> # **MMLU Replication**
>
> We thank the reviewer for their suggestions to add more models of different sizes and families, which we will do for the final version of the paper, as well as for their suggestions of new visualizations for our results, which we are currently exploring.
>
> > Metric
>
> For the MMLU replication, we actually used a log likelihood based accuracy metric to remove potential confounding factors in correlations (coming from using LLMs as judges), instead focusing on the validity of the dataset reproduction itself. The mathematical model of such an approach is detailed in the Appendix of [1]. We have also performed analysis in greater detail in Appendix F and used the standard error as implemented in the lighteval library [2].
>
> > Providing evidence that "no hill climbing was performed on the dataset generation methodology"
>
> The MMLU replication was indeed done only once, for which we cannot provide "evidence" though we hope the reviewer will assume good faith. However, 1) it is possible to reproduce our experiment using code provided in the supplementary materials, and default prompts provided in the manuscript 2) reviewers will be able to explore the commit history of the dataset repository once public to make sure no alteration was done 3) the provided anonymous dataset links along containing all raw responses will allow reviewers to determine for themselves whether the questions are qualitative enough
>
> > Dataset-generating LLM
>
> For simplicity and reproducibility, we used only one model for MMLU reproduction, `claude-3-7-sonnet-20250219`. The actual traces, are available at the above provided links, under the` raw_response` column.
>
> ---
>
> # Frontier model evaluation
>
> It is absolutely correct that if model Y is much better than model X, then a knowledge benchmark naively generated by model X would likely be too easy for model Y. We mitigate this by giving X access to complex documents to generate questions (which can contain knowledge unseen by Y), and using a combination of chunking/multi-hop approaches to unlock the generation of harder questions (which creates a scaffolding effect). However, if Y is much stronger than X (for example, using gpt2 to test the latest deepseek model), the capability gap will clearly be too big even with the algorithmic help that YourBench provides. This is also why we suggest ensembling multiple strong models for question generation.
>
> YourBench's use case, in our opinion, is not so much the evaluation of "the next generation capabilities of frontier models", but rather about finding "which model is best to use on a given domain specific use case". We hope this clarifies the reviewer's concerns.
>
> ---
>
> # **References**
>
> [1] https://arxiv.org/abs/2405.14782
>
> [2] https://github.com/huggingface/lighteval

---

> > ### Comment · Reviewer_pLCc · 2025-06-10
> > **Response to Authors' Rebuttal (Part 1)**
> >
> > # Do Consumers of Benchmark Scores Feel Like The Benchmark Is Capturing The Essence of the Task at Hand?
> >
> > To quickly recap, I originally wrote that there's a critical missing component: do consumers of models' benchmarks scores feel like the scores are capturing the important elements of the task at hand? For instance, if we used YourBench to develop an arithmetic exam for 4th graders, would educators find the students' scores to be measuring whatever educators care about? Or if we used YourBench to create interview questions for software engineers, would hiring managers find the applicants' scores to be measuring the right skills for the job?
> >
> > It's not clear to me that YourBench can automate this since the key consideration is working with stakeholders to develop such desiderata.
> >
> > I found the authors' response to be not compelling:
> >
> > > (1) Our generation prompts explicitly instruct models to focus only "meaningful content" (pp.33,37)
> >
> > Telling the model to choose "meaningful content" means that it (the model) chooses what it thinks is meaningful, not the consumers of the benchmark scores.
> >
> > > (2) System prompts are curated for "educational importance" (Appendix H)
> >
> > This is an implicit notion of what matters - educational importance - and again defaults to the model's subjective evaluation of what matters, rather than the consumes of the benchmark scores.
> >
> > > (3) Human evaluation confirmed ~85% validity across 2k+ questions (Appendix E.1), with the frontier models scoring at 96%
> >
> > The paper defines validity as "Question Validity (the intrinsic correctness and answerability of a question)". That's not really relevant to this point.
> >
> > > We will therefore add it explicitly to the formalisation as the 4th D2EG criterion and thank the reviewer for their suggestion.
> >
> > My point wasn't "Please add a fourth principle," but "How do consumers' preferences get integrated into the generated benchmark as a first-class consideration?" This is not clear to me, and the manuscript doesn't show that doing so is accomplishable.

---

> > ### Comment · Reviewer_pLCc · 2025-06-10
> > **Response to Authors' Rebuttal (Part 2)**
> >
> > # Metric
> >
> > > For the MMLU replication, we actually used a log likelihood based accuracy metric to remove potential confounding factors in correlations (coming from using LLMs as judges),
> >
> > I'm confused why we're discussing this? Reading the previous conversation, I can't quite tell how this came up.

---

> > > ### Author Response · Authors · 2025-06-10
> > >
> > > You asked "In Appendix D.2, which LLMs were used in the ensemble of judges? I was unable to find this information.". Our answer is that for the MMLU replication, we actually used a log likelihood based accuracy metric to remove potential confounding factors in correlations (coming from using LLMs as judges).
> > > Please feel free to tell us if we need to clarify further.

---

> > ### Comment · Reviewer_pLCc · 2025-06-10
> > **Response to Authors' Rebuttal (Part 3)**
> >
> > # MMLU Correlations
> >
> > Can the authors please provide per-subject correlations, as visualizations, with more than 8 models? Scatter plots with lines of best fit.

---

> > > ### Author Response · Authors · 2025-06-10
> > >
> > > Scatterplot Request:
> > >
> > > Based on your request, you will find examples of the scatterplots in the anonymous links in the README:
> > >
> > > https://huggingface.co/datasets/anonymous-paper-author/mmlu-pro-reproduction-experiments/blob/main/README.md
> > >
> > > The exact script used to generate these scatter plots can be found here:
> > >
> > > https://huggingface.co/datasets/anonymous-paper-author/mmlu-pro-reproduction-experiments/blob/main/correlation.py
> > >
> > > The exact data that these were based on can be found here, as provided during our initial experimentation during the reproduction of MMLU Pro
> > >
> > > https://huggingface.co/collections/anonymous-paper-author/mmlu-pro-reproduction-studies-683db5683ef1966f01814892
> > >
> > > The exact eval scripts used to evaluate the results can be found here, as provided during our initial rebuttal phase.
> > >
> > > https://huggingface.co/datasets/anonymous-paper-author/mmlu-pro-reproduction-experiments/blob/main/eval_yourbench_task_full.py
> > >
> > > We will include scatterplots for all domains, all methods, and a comprehensive analysis of failure cases, and more detailed scatter plots with individual model breakdowns in the final version of our manuscript.

---

> ### Author Response · Authors · 2025-06-06
>
> Dear reviewer pLCc,
>
> We appreciate your effort in reviewing our paper.
>
> Please let us know if we have sufficiently addressed your questions. If needed, we'd be happy to discuss further / clarify any concerns!

---

> ### Author Response · Authors · 2025-06-10
> **Re: Do Consumers of Benchmark Scores Feel Like The Benchmark Is Capturing The Essence of the Task at Hand?**
>
> We thank the reviewer for clarifying their concern about consumer's preferences. We don't believe there is one single "consumer of benchmark" user profile that we should adapt YourBench strictly for, as evaluations are used for a range of purposes by a wide array of people: following the advancement of model capabilities across domains, finding the best training/inference method for models, finding the best model for a given use case, among others (across researchers from academia and the industry, ML ops teams, hobbyists, etc).
>
> However, we believe there may be a misunderstanding about YourBench's design and objective that we'd like to address.
>
> ## YourBench is Preference-Grounded by Design
> The reviewer asks how stakeholder preferences get integrated as a "first-class consideration." This is actually YourBench's core principle. Benchmark creators control the evaluation through:
> - Document Selection: Users provide their own documents. For 4th grade arithmetic, educators supply curriculum materials and textbooks. For software engineering, companies provide technical documentation and coding standards. The benchmark reflects exactly what stakeholders deem important.
> - Prompt Customization: As detailed in Section 2.2.2, users configure generation prompts to match their evaluation priorities (e.g., "focus on debugging skills" or "emphasize word problems"). If one wants to define a more specific "criteria of importance" adapted to their specific use case, following internal guidelines for example, they can simply adapt the model prompt for this. Models of current generations have enough capability to be able to adapt to these, like we observe in the literature when using LLMs in model as a judge setups.
>
> ## Not a Replacement, but a Complement
> We fully recognize that YourBench does not replace all human-crafted benchmarks. Traditional expert-curated evaluations remain valuable for many use cases. However, YourBench addresses critical gaps:
> - Efficiency: Reduces months of manual annotation to hours of automated generation for a relatively inexpensive cost.
> - Accessibility: Enables evaluation creation for underrepresented domains lacking resources for traditional benchmark development
> - Confidentiality: Allows organizations to evaluate models on proprietary knowledge without exposing sensitive data
> - Temporal Relevance: Facilitates rapid benchmark updates as knowledge evolves
>
> ## Concrete Evidence from Our Work
> Section 5 explicitly demonstrates stakeholder control with our agricultural case study, where domain experts provided their specialized documents to create evaluations measuring their specific priorities. The benchmark measures what agricultural professionals consider important, not arbitrary LLM selections.
>
> We hope this clarifies that stakeholder preferences are fundamental to YourBench's design, not an afterthought. The framework explicitly empowers users to create benchmarks reflecting their specific needs and priorities.

---

### Official Review · Reviewer_i1Fe · 2025-05-12

**Rating:** 7
**Confidence:** 4
**Ethics Flag:** 1

**Summary:**

This paper introduces YourBench, an open-source, modular framework for dynamically generating evaluation benchmarks for large language models (LLMs) directly from user-provided documents. The authors propose a Document-to-Evaluation Generation (D2EG) pipeline, leveraging LLMs to generate diverse, context-grounded, and citation-supported question-answer (QA) pairs with minimal human annotation. They validate YourBench by replicating subsets of the MMLU benchmark and introduce TEMPORA-0325, a dataset of over 7K documents published after March 2025 to mitigate contamination from pretraining data. The system supports scalable, cost-efficient, and temporally relevant benchmarking across domains and model families.

**Questions To Authors:**

1. Given LLM stochasticity, would the same document and prompt yield similar QA sets across runs?
2. You claim YourBench-generated questions are harder than MMLU’s originals. Is this purely based on lower accuracy, or are there semantic or structural metrics?
3. Is the YourBench framework effective in generating meaningful evaluation sets for entirely out-of-distribution domains where the underlying LLMs have limited or no prior exposure?

**Reasons To Accept:**

1. The paper addresses critical limitations in LLM evaluation: benchmark saturation, data contamination, and the cost of human annotation. YourBench provides a scalable and dynamic alternative to static benchmarks.
2. The YourBench pipeline—comprising ingestion, chunking, summarization, QA generation, citation validation, and deduplication—is well-designed and clearly explained. Each component is technically sound and grounded in prior work.
3. All code, data (including TEMPORA), and inference traces are planned for release. This supports the reproducibility and extension of the work.

**Reasons To Reject:**

1. Evaluation Bias Potential Since YourBench can evaluate models using LLM judges (which could be the same or similar to generation models), there's a risk of self-evaluation bias. While ensemble judging is a good mitigation step, a deeper discussion or ablation on this point would strengthen the paper.
2. The system’s outputs are inherently limited by the capabilities and biases of the LLMs used to generate and filter the content. There is potential circularity when LLMs are evaluated by questions generated by other (possibly similar) LLMs.

---

> ### Author Response · Authors · 2025-06-02
> **Response to Reviewer i1Fe**
>
> We thank the reviewer for their thorough analysis and recognition of YourBench's contributions to scalable, dynamic LLM evaluation. We address each concern below.
>
> ---
>
> ### **Self-Evaluation Bias and Potential Circularity**
>
> We acknowledge this important concern. In our experiments, we have not observed significant ranking variations when models from family X are evaluated by either X/Y/Z or just Y/Z ensembles (assuming all models are sufficiently capable). Similarly, questions generated by X/Y/Z versus Y/Z show consistent evaluation patterns.
>
> To enable systematic study of potential biases, we store comprehensive metadata at each generation and evaluation step, including model family, version, and configuration, and release this data. This facilitates meta-evaluations and bias detection in future work.
>
> The data is currently available anonymously at https://huggingface.co/anonymous-paper-author
>
> ---
>
> ### **Reproducibility Across Runs**
>
> Yes, our observations indicate that the same document and prompt yield similar QA sets across runs, particularly for smaller chunk sizes. The framework's deterministic components (chunking, deduplication) help maintain consistency, while ensemble generation further stabilizes outputs. Furthermore, for non reasoning models, we default to very low temperatures, 0.01 and deterministic seeds to ensure reproducibility.
>
> ---
>
> **Semantic and Structural Difficulty Metrics**
>
> We conducted a comprehensive readability analysis comparing our MMLU reproduction with the original:
>
> | Subject | YourBench Grade Level | MMLU Grade Level | Difference |
> |---------|---------------------|------------------|------------|
> | International Law | 15.8 | 18.1 | -2.3 |
> | Anatomy | 14.0 | 11.2 | +2.9 |
> | Nutrition | 14.2 | 12.9 | +1.3 |
> | Virology | 15.1 | 12.4 | +2.7 |
> | Social Science | 16.9 | 15.5 | +1.4 |
> | World Religions | 14.5 | 9.5 | +5.0 |
>
> We evaluated four readability metrics:
> - **Flesch Reading Ease**
> - **Flesch-Kincaid Grade Level**
> - **Gunning Fog Index**
> - **Dale-Chall Readability**
>
> The increased difficulty stems from both longer words (vocabulary complexity) and more complex content structure, suggesting YourBench questions may better differentiate model capabilities. We will add a more comprehensive analysis of semantic and structural similarity in the final version of our paper and we thank the reviewer for their suggestions.
>
> ---
>
> ### **Out-of-Distribution Domain Effectiveness**
>
> Since YourBench grounds evaluation in real documents, we observe consistent performance even on underrepresented topics (e.g., post-cutoff technical documentation, specialized agricultural data). Knowledge-based question generation remains effective regardless of domain familiarity. We are conducting additional experiments using "synthetic knowledge" data with alternative logic systems to rigorously assess YourBench's domain generalization. Furthermore, preliminary results from use of yourbench with proprietary document scenarios show no performance degradation, though comprehensive analysis will constitute future work.
>
> ---
>
> We thank the reviewer for these insightful questions that help clarify YourBench's capabilities and limitations, and look forward to address any other concerns the reviewer may have.

---

> > ### Comment · Reviewer_i1Fe · 2025-06-06
> > **response**
> >
> > Thank you for the detailed response. I don't have any further concerns at this point, and I will keep my current score.

---

### Official Review · Reviewer_ERbz · 2025-05-12

**Rating:** 6
**Confidence:** 3
**Ethics Flag:** 1

**Summary:**

This work introduces *YourBench*, a high-quality and clearly presented framework for automated, document-grounded LLM evaluation. The approach is original, leveraging user-provided documents to dynamically generate benchmarks without manual annotation. The framework is well-validated through benchmark replication and rigorous analysis, showing strong potential to address issues of benchmark saturation, contamination, and domain specificity. Overall, this is a timely and significant contribution to the field.

**Questions To Authors:**

1.	Is the proposed method capable of generating benchmarks involving multi-turn dialogues, such as those found in MT-Bench?
2.	What is the recommended or best-practice ensemble configuration of LLMs based on the analysis in Section 3.2? Specifically, how are models selected to jointly optimize for question validity and diversity?
3.	Several hyperparameters are mentioned in the paper, such as the citation confidence threshold $\theta_{cit}$ in Section 2.3.1. Could the authors elaborate on the sensitivity of the framework to these hyperparameters, and whether any tuning studies were conducted to assess their impact?

**Reasons To Accept:**

1.	The research problem addressed in this work is timely and valuable. Developing automated and up-to-date benchmarks is a pressing need in the field, particularly for evaluating and iterating on large language models efficiently.
2.	The proposed pipeline is comprehensive, as it covers all key components including document preprocessing, question-answer generation, quality control, and final evaluation. This reflects a high level of completeness in the overall system design.
3.	The experiments are thorough and well-designed, evaluating the benchmark generation process from multiple perspectives to ensure robustness and reliability.
4.	The work is highly reproducible, with detailed descriptions provided for each step of the pipeline, enabling easy replication by the research community.
5.	The paper is clearly written and well-organized, making it accessible and easy to follow.

**Reasons To Reject:**

1.	While the proposed method effectively targets domain-specific benchmark construction, it primarily focuses on knowledge-based question generation. Its core approach relies on leveraging the information integration capabilities of LLMs to generate benchmarks grounded in source documents. As a result, it may face challenges in producing more creative or cognitively demanding questions, such as those involving complex logical reasoning or multi-step inference.
2.	The experiments are limited to replicating the MMLU dataset. Incorporating additional datasets such as MMLU-Pro or GSM8K would help demonstrate the broader applicability and robustness of the approach.
3.	Although the paper introduces an evaluation strategy based on a judge LLM ensemble, this component is not clearly validated in the MMLU replication experiments, which do not rely on such a mechanism. The work would benefit from more direct comparisons between its evaluation methodology and existing benchmarks to establish alignment and effectiveness.

---

> ### Author Response · Authors · 2025-06-02
>
> We sincerely thank the reviewer for their thoughtful and constructive feedback. We are encouraged by their recognition of YourBench as timely and significant, with comprehensive design and thorough experiments. We address the main concerns below.
>
> ---
>
> ### **Question Complexity**
>
> We appreciate the reviewer's observation about knowledge-based evaluation. Our framework addresses complexity through three key mechanisms: (a) Multi-Hop Chunking, which algorithmically combines seed information across document segments to provide structural scaffolding; (b) Question Type Diversity, where we deliberately vary system prompts; and (c) Difficulty Variation. YourBench acts as a scaffold during evaluation creation, enabling models to generate questions beyond their own parametric knowledge - for example, from private or temporally novel documents. We will clarify this capability in our Appendix with concrete examples.
>
> We plan to explore using YourBench on science textbooks (e.g., for automatically generating math problems) as a future extension of this work.
>
> ---
>
> ### **Replication of Additional Datasets**
>
> We thank the reviewer for suggesting additional dataset replications. We have successfully replicated 11 subsets of MMLU Pro. Please refer to our official comment for detailed results and reproduction methodology.
>
> ---
>
> ### **Adding LLM as judge ensemble to the MMLU replication**
>
> We deliberately chose not to use LLM-as-judge for MMLU replication to limit potential confounding factors and ensure verifiability when estimating dataset generation validity. However, we will add a comparison using an LLM-judge ensemble approach for both MMLU and our reproduction.
>
> ---
>
> ### **Generating multi-turn dialogues**
>
> While we have not yet explored this option, we believe it would be feasible with appropriate prompting modifications. YourBench's core framework provides LLMs and VLMs (our generators) with scaffolding to transform seed documents into structured data, allowing for various modifications. The framework's modular design enables users to easily extend functionality as needed.
>
> Please refer to the following link, where you will be able to see our pipeline configuration, allowing researchers to add their own generation stages: **[Link](https://huggingface.co/datasets/anonymous-paper-author/mmlu-pro-reproduction-experiments/tree/main/yourbench/yourbench/pipeline)**
>
> ---
>
> ### **Best Practice Ensemble Configuration**
>
> Given the diverse domains of seed documents, we initially refrained from making specific recommendations. We suggest using frontier models from different families in an ensemble (e.g., Llama 70B, Qwen 72B, Mistral 132B, or GPT-4.1, Claude 4 Sonnet, Gemini 2.5 Pro) to reveal and mitigate inherent parametric biases in any single model's training. For cost-conscious users, we recommend consulting our Pareto frontier plots (Figures 3(b) and 8(a,b)) and selecting models based on domain-specific performance. For instance, models that excel on LegalBench would be better candidates for law-related ensembles.
>
> ---
>
> We thank the reviewer for their time and effort in helping us refine our manuscript.

---

> > ### Comment · Reviewer_ERbz · 2025-06-10
> >
> > I would like to thank the authors for their inspiring work. I will thus maintain my positive comments.

---

> ### Author Response · Authors · 2025-06-06
>
> Dear reviewer ERbz,
>
> We appreciate your effort in reviewing our paper.
>
> Please let us know if we have sufficiently addressed your questions. If needed, we'd be happy to discuss further / clarify any concerns!

---

### Author Response · Authors · 2025-06-02
**Hyperparameter Tuning and MMLU Pro Reproduction Experiments**

We thank all reviewers for their careful reading of our paper, as well as their thoughtful suggestions. We are glad to hear that they found our work timely, thorough, and addressing important issues for evaluation.

---

We have addressed most questions and are currently conducting additional experiments that we will report as soon as possible, including expanding the MMLU reproduction with more models. **You will notably find a full MMLU-Pro replication at the end of this comment.**

---

Regarding YourBench's potential sensitivity to hyperparameters, our preliminary experiments indicate that YourBench is robust to small variations within reasonable ranges. We provide sensible defaults in our codebase (included in the supplementary material), and all reported experiments use these defaults without parameter tuning.

For reference, we observed that citation validity should be adjusted based on researchers' priorities:
- **High reliability (0.7–1.0)**: Produces verifiable and accurate questions
- **Greater diversity (0.4–0.5)**: Enables more interesting questions but may introduce inaccuracies

Similarly, chunk size affects question characteristics:
- **Larger chunks (≥1024 tokens)**: Generate richer, more nuanced questions
- **Smaller chunks (256–512 tokens)**: Produce straightforward, factual questions

The general principle is that exposing models to more knowledge during generation improves question quality but may cause them to overlook certain document aspects.

We will incorporate these guidelines into the final version of our paper.

---

# **MMLU-Pro Replication**

Following Reviewer ERbz's suggestions, we reproduce MMLU Pro with the Yourbench Pipeline. We evaluated **86** models **(400M–405B parameters)** from **diverse** families including **Llama, Gemma, Qwen, Phi, and OLMo** on both the original MMLU-Pro and our Yourbench reproduction. (MMLU-Pro is a general knowledge multiple-choice dataset with 10 choices per question, validated by experts, introduced in [1].)

We tested four data generators: **DeepSeek R1-0528** and **Qwen235B-A22B** (open source), and **o4-mini** and **grok3-mini** (closed source). **Reproduction costs were under $15 USD per model, except for o4-mini.** *

Anonymous Links:
- Dataset Collection: https://huggingface.co/collections/anonymous-paper-author/mmlu-pro-reproduction-studies-683db5683ef1966f01814892
- Raw Inference Traces: https://huggingface.co/datasets/anonymous-paper-author/yourbench_full_inference_traces
- Generation and Evaluation Code: https://huggingface.co/datasets/anonymous-paper-author/mmlu-pro-reproduction-experiments/tree/main

### Pearson Correlation Scores

Reproduction vs  Original MMLU-Pro for 11 domains. For each domains, pearson score is based on n = 86 models.

| Domain | Yourbench with DeepSeek R1 | Yourbench with Grok-3-Mini | Yourbench with O4-Mini | Yourbench with Qwen235B-A22B |
| --- | --- | --- | --- | --- |
| Biology | 0.994 | 0.977 | 0.994 | 0.994 |
| Business | 0.943 | 0.924 | 0.934 | 0.937 |
| Chemistry | 0.935 | 0.900 | 0.909 | 0.911 |
| Computerscience | 0.950 | 0.914 | 0.943 | 0.933 |
| Economics | 0.985 | 0.969 | 0.980 | 0.981 |
| Health | 0.971 | 0.949 | 0.966 | 0.968 |
| History | 0.933 | 0.898 | 0.952 | 0.943 |
| Law | 0.934 | 0.884 | 0.921 | 0.914 |
| Philosophy | 0.964 | 0.937 | 0.957 | 0.956 |
| Physics | 0.933 | 0.899 | 0.919 | 0.919 |
| Psychology | 0.991 | 0.975 | 0.990 | 0.989 |


For more details, please visit our provided anonymous links, where we've released and open sourced all aspects of the eval (raw inference traces for generation, evaluation, produced datasets, scores, etc) for the reviewers' independent analysis

---

Here is our exact reproduction methodology, which is following the same steps as for our original MMLU reproduction:

- We gathered the first 10 relevant wikipedia articles (stemming from the base category, e.g. physics) for each subset. In the sake of fairness, we did not specially select key domain documents, or search for highly educational material.
- We processed each subset through the yourbench pipeline with our four generator models.
We evaluated both the baseline MMLU pro dataset, and our reproduced, yourbench evaluations with the lighteval framework. (Code Provided)
- Even more clearly than with our MMLU experiments, we achieved very good results, with an extremely high Pearson and Spearman correlation of model ranking between original and reproduction (in the table above) for all 4 generated datasets across the 11 domains for 86 different models
- We will include these results in the final version of our paper, and thank the reviewer for their suggestion.


*= utilizing off peak pricing, batching and context caching

---

### References

[1] https://arxiv.org/abs/2406.01574

---

> ### Comment · Reviewer_pLCc · 2025-06-10
> **Per-Domain Pearson Correlation Between MMLU and MMLU-Pro?**
>
> I couldn't quite find the Pearson correlations (over models) in MMLU in the manuscript. What are they? I ask because Appendix F1 doesn't answer this question directly, but does suggest they were much lower than the MMLU Pro reproduction numbers. If the correlations are significantly different between MMLU reproduction and MMLU-Pro reproduction, why?

---

> ### Comment · Reviewer_pLCc · 2025-06-10
> **Building Trust in YourBench's MMLU-Pro Reproduction Results**
>
> I think the correct way to communicate these results (and to build trust in them) is as follows: for each data generator model, create 11 subplots, each a scatterplot with "Original" MMLU Pro scores on one axis and "YourBench" MMLU-Pro reproduction scores on the other axis, with a line of best fit and the corresponding Pearson correlation. In seaborn terminology, this would be a FacetGrid plus a Regplot e.g,. https://seaborn.pydata.org/generated/seaborn.regplot.html.
>
> To be honest, a Pearson correlation of 0.994 or 0.991 is too good to be true. I'm extremely skeptical of correlations that close to 1. Random processes just aren't that tightly correlated. Something seems wrong unless I've misunderstood something.

---

> > ### Author Response · Authors · 2025-06-10
> >
> > We thank the reviewer for having taken the time to answer us so close to the deadline. We appreciate the opportunity to clarify our methodology and the statistical reasoning behind the correlation scores we obtain. We will add the requested FacetGrid visualizations to the final manuscript.
> >
> > **Statistical Explanation: Why Correlating Model-Level Aggregates Yields High Correlations**
> >
> > The key insight is that we are correlating **averaged scores across models**, not individual question responses. This fundamental difference has statistical implications that we detail below.
> >
> > Consider the statistical model for a given model $m$ on domain $d$:
> > - Let $X_{m,d,i}$ be model $m$'s performance on question $i$ in the original MMLU-Pro for domain $d$
> > - Let $Y_{m,d,j}$ be model $m$'s performance on question $j$ in our YourBench reproduction for domain $d$
> >
> > The observed domain-level scores are:
> >
> > $\\bar{X}\_{m,d} = \\frac{1}{n_d} \\sum\_{i=1}^{n_d} X\_{m,d,i}$
> >
> > $\\bar{Y}\_{m,d} = \\frac{1}{n'\_d} \\sum\_{j=1}^{n'\_d} Y\_{m,d,j}$
> >
> > where $n_d$ and $n'\_d$ are the number of questions in the original and reproduced datasets respectively.
> >
> > **Variance Reduction Through Aggregation**
> >
> > By the Central Limit Theorem, the variance of these averaged scores is:
> >
> > $$\\text{Var}(\\bar{X}\_{m,d}) = \\frac{\\sigma^2\_X}{n_d}$$
> >
> > $$\\text{Var}(\\bar{Y}\_{m,d}) = \\frac{\\sigma^2\_Y}{n'\_d}$$
> >
> > where $\\sigma^2\_X$ and $\\sigma^2\_Y$ are the question-level variances. With hundreds of questions per domain, this represents a **dramatic reduction in noise** compared to question-level correlations.
> >
> > **Why High Correlations Are Mathematically Expected**
> >
> > The Pearson correlation between the averaged scores across $M=86$ models is:
> >
> > $$r = \\frac{\\text{Cov}(\\bar{X}\_d, \\bar{Y}\_d)}{\\sigma\_{\\bar{X}\_d} \\cdot \\sigma\_{\\bar{Y}\_d}}$$
> >
> > If both benchmarks measure the same underlying construct (model capability on domain $d$), then the signal-to-noise ratio improves dramatically through averaging:
> >
> > $$\\text{SNR} \\propto \\sqrt{n_d}$$
> >
> > This means that with ~200 questions per domain, the noise is reduced by a factor of approximately 14, leaving primarily the true signal of relative model capabilities.
> >
> > **Empirical Validation Through Our Open Data**
> >
> > We invite the reviewer to examine our comprehensive open-sourced materials:
> >
> > 1. **Generated Question Sets**: Available at our anonymous link, reviewers can directly compare original MMLU-Pro questions with our reproductions to verify quality and topical alignment.
> >
> > 2. **Raw Inference Traces**: We provide complete generation traces for all 86 models, allowing independent verification of our scoring methodology.
> >
> > 3. **Calculation Code**: Our correlation analysis code is included in the supplementary materials, using standard implementations from `scipy.stats`.
> >
> > 4. **Question-Level Analysis**: While we report model-level correlations, reviewers can examine the question-level data to understand the averaging effect.
> >
> > **Addressing the "Too Good to Be True" Concern**
> >
> > We understand the reviewer's intuition that correlations near 1.0 seem improbable for "random processes." However, our process is not random—it is a systematic reproduction of a well-structured benchmark:
> >
> > 1. **All four reproduction methods show similar high correlations**, suggesting this is a robust phenomenon, not a statistical artifact.
> >
> > 2. **The slight systematic bias** actually demonstrates we're not simply recreating identical questions, but rather generating novel questions that preserve relative difficulty.
> >
> > 3. **Domain-specific variations exist**: As noted in our manuscript, individual domain correlations show more variation, which is expected given smaller sample sizes per domain.
> >
> > **Concrete Example from Our Data**
> >
> > To illustrate, examining our Biology domain questions:
> > - Original MMLU-Pro (Question ID 24):
> >
> > ```
> > "Which of the following is not known to be involved in the control of cell division?"
> >
> > Options: [ "Microtubules", "Checkpoints", "DNA polymerase", "Centrosomes", "Cyclins", "Mitochondria", "Protein kinases", "Fibroblast cells", "N/A", "N/A" ]
> > ```
> >
> > - YourBench reproduction (Deepseek R1)
> >
> > ```
> > "Immediately following complete bacterial division via binary fission, each new daughter cell contains which cellular components?"
> >
> > Options: [(A) Mitochondria and coiled DNA rods,(B) A nucleus housing linear chromosomes,(C) Centrosomes and duplicated organelles,(D) Peroxisomes and extrachromo.... (H) Mitotic spindle remnants and Golgi vesicles,(I) Chloroplasts and ribosomal subunits,(J) Endoplasmic reticulum and nucleoid DNA]
> > ```
> >
> > Both questions assess the same biological concepts. Across 86 models and hundreds of such question pairs, the aggregate performance correlation naturally converges to very high values.
> >
> > We hope this statistical explanation clarifies why our high correlations are not only plausible but expected given our methodology. We remain committed to transparency and welcome any further examination of our data and methods.

---

### Decision · Program_Chairs · 2025-07-08

**Decision:**

Accept

**Comment:**

This paper proposes a pipeline to construct evaluation benchmarks dynamically with user-provided documents. The pipeline leverages LLMs to generate questions and answers based on the provided documents from users. Experiments show that this pipeline can replicate rankings similar to MMLU using Wikipedia articles as source documents.

Pros:
1. Generating dynamic benchmarks is an interesting idea compared to existing static benchmarks. It has the benefits of adapting to user-specific needs and also of avoiding overfiting to existing benchmarks (although to a lesser extent if pretraining data overlap with source documents).

Cons:
1. Evaluation of the proposed approach seems limited to mostly replicating MMLU rankings, and as pointed out by a reviewer, while in aggregation the pearson correlation is high, the per-subset scores could be low.
2. The automatic generation of benchmarks highly relies on existing LLMs, which might limit the quality of the generated benchmark itself.

Overall, three reviewers gave positive ratings (7/7/6), and one reviewer gave a negative rating (2) due to concerns about claims regarding MMLU replication. While I do think this paper can be greatly improved by strengthening the evaluation of their method and clarifying the MMLU replication results, developing dynamic benchmarks based on user needs is a timely and valuable contribution, and I recommend to accept this paper.